



# NH₃ emissions from large point sources derived from CrIS and IASI satellite observations

Enrico Dammers[1], Chris A. McLinden[1], Debora Griffin[1], Mark W. Shephard[1], Shelley Van Der Graaf[2], Erik Lutsch[3], Martijn Schaap[4], Yonatan Gainairu-Matz[1], Vitali Fioletov[1], Martin Van Damme[5], Simon Whitburn[5], Lieven Clarisse[5], Karen Cady-Pereira[6], Cathy Clerbaux[5,7], Pierre Francois Coheur[5], and Jan Willem Erisman[2,8]

[1]Environment and Climate Change Canada, Toronto, Ontario, Canada
[2]Cluster Earth and Climate, Department of Earth Sciences, Vrije Universiteit Amsterdam, Amsterdam, the Netherlands
[3]Department of Physics, University of Toronto, Toronto, Ontario, Canada
[4]TNO, Climate Air and Sustainability, Utrecht, The Netherlands
[5]Université libre de Bruxelles (ULB), Service de Chimie Quantique et Photophysique, Atmospheric Spectroscopy, Brussels, Belgium
[6]Atmospheric and Environmental Research (AER), Lexington, MA, United States of America
[7]LATMOS/IPSL, Sorbonne Université, UVSQ, CNRS, Paris, France
[8]Louis Bolk Institute, Driebergen, the Netherlands

**Correspondence:** Enrico Dammers (enrico.dammers@canada.ca / enrico.dammers@gmail.com)

**Abstract.** Ammonia ($NH_3$) is an essential reactive nitrogen species in the biosphere and through its use in agriculture in the form of fertilizer important for sustaining human kind. The current emission levels however, are up to four times higher than in the previous century and continue to grow with uncertain consequences to human health and the environment. While $NH_3$ at its current levels is a hazard to the environmental and human health the atmospheric budget is still highly uncertain, which

5   is a product of an overall lack of measurements. The capability to measure $NH_3$ with satellites has opened up new ways to study the atmospheric $NH_3$ budget. In this study we present the first estimates of $NH_3$ emissions, lifetimes, and plume widths from large (>∼5 kt yr$^{-1}$) agricultural and industrial point sources from CrIS satellite observations across the globe with a consistent methodology. The same methodology is also applied to the IASI (A and B) satellite observations and we show that the satellites typically provide comparable results that are within the uncertainty of the estimates. The computed $NH_3$ lifetime

10  for large point sources is on average 2.35±1.16 hours. For the 249 sources with emission levels detectable by the CrIS satellite, there are currently 55 locations missing (or underestimated by more than an order of magnitude) from the current HTAPv2 emission inventory, and only 72 locations with emissions within a factor 2 compared to the inventories. We find a total of 5622 kt yr$^{-1}$, for the sources analyzed in this study, which is equivalent to a factor ∼2.5 between the CrIS estimated and HTAPv2 emissions. Furthermore, the study shows that it is possible to accurately detect short and long-term changes in emissions, demonstrating the possibility of using satellite observed $NH_3$ to constrain emission inventories.



# 1 Introduction

Ammonia ($NH_3$) is one of the most important reactive nitrogen species in the biosphere and essential for sustaining human kind through its use in agriculture in the form of fertilizer. However, at the current emission levels, which are up to four times higher than in the previous century, and which continue to grow with uncertain consequences (Holland et al., 1999; Fowler
et al., 2013; Battye et al., 2017), which raises major environmental and health challenges (Sutton et al., 2009; Rockstrom et al., 2009). Through deposition $NH_3$ can cause eutrophication and acidification (Erisman et al., 2007; EEA-European Environment Agency, 2014), which can lead to a reduction of biodiversity in vulnerable ecosystems (Bobbink et al., 2010). On the other hand, through fertilization of ecosystems, deposited $NH_3$ and other reactive nitrogen species play an important role in the sequestration of carbon dioxide (Oren et al., 2001; de Vries et al., 2014). $NH_3$ reacts rapidly with acidic gases, forming
ammonium nitrate and ammonium sulfate, which combined total up to $50\%$ of the mass of fine particulate matter (Schaap et al., 2004; Seinfeld and Pandis, 2012). Particulate matter is shown to have impact on human health, shortening human life expectancy and affecting pregnancy outcomes (Pope III et al., 2002, 2009; Stieb et al., 2012; Lelieveld et al., 2015; Giannakis et al., 2019). Furthermore, particulate matter impacts global climate change directly through the change in radiative forcing and indirectly through its effects on cloud formation (Adams et al., 2001; Myhre et al., 2013).

While measures to reduce $NO_x$ and $SO_2$ emissions have had a significant and measurable effect (Krotkov et al., 2016; Lamsal et al., 2015), emission control measures for $NH_3$ have not sufficiently been implemented, and the efficiency of the proposed measures is uncertain. Several studies (Erisman and Schaap, 2004; Xu et al., 2017) have shown the importance of similar emission controls for $NH_3$. Recently though there has been significant debate on the quality of both modelled emissions and observed concentrations, as modelled long-term trends do not match observed trends. Over the last decade, several studies
have focused on solving this discrepancy, but have been unsuccessful to date (Yao and Zhang, 2016; Van Zanten et al., 2017; Wichink Kruit et al., 2017). One of the problems is the lack of dense and precise measurement networks. $NH_3$ concentrations are highly variable in time and space, as shown by in-situ and aircraft observations. With estimated lifetimes of a few hours up to a few days, a wide distribution of measurements, both spatially and temporally, are needed to understand the processes and infer a representative atmospheric budget. Globally there are only a few measurements networks, with few consisting of
more than a small number of locations, and even fewer with a consistent measurement record or providing vertical profiles (Erisman et al., 1988; Dammers et al., 2017a; Tevlin et al., 2017; Zhang et al., 2018b). This makes it hard to study and resolve the discrepancy in the trends. Furthermore, most measurements are performed at a coarse temporal resolution (up to monthly averages) while most of the atmospheric processes, such as emissions, deposition and chemical conversion to particulate matter ($PM_{2.5}$), take place on timescales of minutes to hours.

The availability of satellite observations in recent years has provided global coverage of $NH_3$, opening up new ways to study the atmospheric $NH_3$ budget. Instruments such as the Tropospheric Emission Spectrometer (TES, Beer et al. (2008); Shephard et al. (2011)), the Atmospheric Infrared Sounder (AIRS, Warner et al. (2016)), the Infrared Atmospheric Sounding Interferometer (IASI, Clarisse et al. (2009); Coheur et al. (2009); Van Damme et al. (2014a)), and the Cross-track Infrared



Sounder (CrIS, Shephard and Cady-Pereira (2015)) have shown that it is possible to retrieve $NH_3$ from spectra obtained by satellites. These satellite $NH_3$ observations have greatly improved our understanding of its distribution. Studies showed the global distribution of $NH_3$ measured one or two times per day (Van Damme et al., 2014a; Shephard and Cady-Pereira, 2015) can reveal seasonal cycles and distributions for regions where measurements are commonly sparse or unavailable (Warner et al.,

2017; Van Damme et al., 2015; Whitburn et al., 2015, 2016b). Several studies have compared satellite measured to modeled $NH_3$ concentrations showing an overall underestimation of the modelled concentrations in North America (Heald et al., 2012; Nowak et al., 2012; Zhu et al., 2013; Schiferl et al., 2014, 2016; Lonsdale et al., 2017), Europe (Van Damme et al., 2014b; Wichink Kruit et al., 2017) and China (Zhang et al., 2018a) likely due to an underestimation of the emissions. The atmospheric lifetime of $NH_3$ is highly uncertain as most of the commonly used ground-based instruments have a temporal resolution that

is too coarse to resolve it. There are only a few estimates reported in literature, ranging from several hours up to 2 days, either obtained by analysing short or long range transport of fire plumes (Yokelson et al., 2009; R'Honi et al., 2013; Whitburn et al., 2015; Lutsch et al., 2016; Adams et al., 2019), or through model estimates, such as by Dentener and Crutzen (1994).

    Satellite observations have been successfully used for direct estimates of emissions and lifetimes of several species (e.g. $SO_2$, $NO_x$, $CO_2$, CO, and $NH_3$) for single anthropogenic point sources (Martin, 2008; Streets et al., 2013; Fioletov et al.,

2015; Nassar et al., 2017), biomass burning (Mebust et al., 2011; Castellanos et al., 2015; Whitburn et al., 2015; Adams et al., 2019) as well as for estimating emissions for multiple sources at the same time (Fioletov et al., 2017). A multitude of approaches exist: simple box models (Jacob, 1999; Hickman et al., 2018; Van Damme et al., 2018), methods that take the wind direction into account (Beirle et al., 2011; Pommier et al., 2013), and more complete methods that account for diffusion and plume shape (Fioletov et al., 2011; de Foy et al., 2014; McLinden et al., 2016; Nassar et al., 2017). In the case of $NH_3$,

there are several studies that report emission estimates based on satellite observations (R'Honi et al., 2013; Whitburn et al., 2015, 2016b; Van Damme et al., 2018; Hickman et al., 2018; Adams et al., 2019). However, due to the low signal-to-noise ratio of single observations, and the relatively coarse spatial resolution, only a few studies have used direct emissions estimates from satellite observations (e.g. Van Damme et al. (2018); Hickman et al. (2018); Adams et al. (2019)). Other studies have commonly used model inversions in various forms (Zhu et al., 2013; Zhang et al., 2018a). Independent of the methods used,

most studies indicate a regional and national underestimation of up to several orders of magnitude for both anthropogenic and natural emissions in most current inventories.

    Here we derive and compare $NH_3$ emissions and, for the first time, lifetimes of globally distributed industrial and agricultural emission sources based on the independent observations of $NH_3$ by CrIS and IASI . We use the complete datasets of IASI-A, IASI-B and CrIS to give estimates for both the 2008-2017 (IASI-A) and 2013-2017 (CrIS, IASI-A and -B) intervals. We show

that all instruments provide comparable emission estimates and similar inter-annual variability. In section 2 we describe the datasets and the methods, and estimate the uncertainty of the method used in this study. In section 3 we describe the results, starting with the fitted lifetime and the plume spread ($\sigma$), which will be used to constrain the final emission fits given in Section





3.2. Furthermore, Section 3.3 describes the temporal variations in the emission estimates. Finally, in Section 4 we summarize and discuss the results.





## 2 Datasets and method

### 2.1 Satellite Products

#### 2.1.1 CrIS

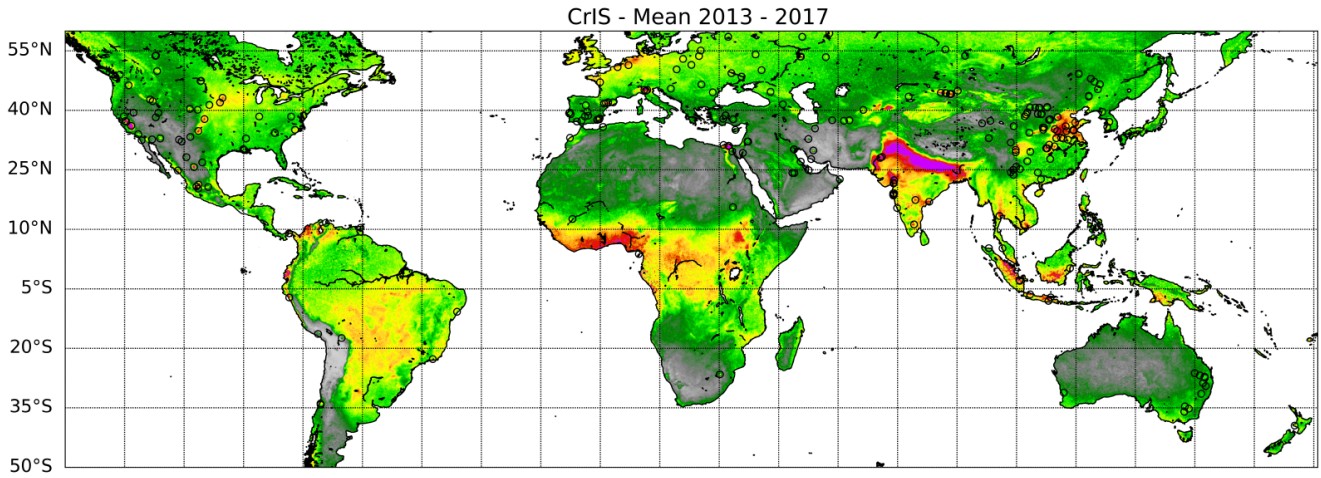

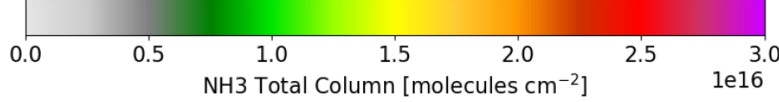

**Figure 1.** CrIS NH$_3$ 5 year mean [2013-2017] total column [molecules cm$^{-2}$] distribution at 0.05° x 0.05° [longitude, latitude] resolution, with the source locations of successful estimates shown by black circles centered around the source locations.

The CrIS instrument is on board of the Suomi-NPP platform launched in October 2011. It orbits the Earth in a sun-

5 synchronous orbit, with twice daily global coverage, crossing the equator at 13:30 and 1:30 local solar time. The instrument covers a wide swath of up to 2200 km width with circular pixels that have a nadir diameter of 14km at nadir. In this study we use version 1.5 data of the CrIS-Fast Physical Retrieval (FPR)-NH$_3$ product (see Shephard and Cady-Pereira (2015) and Shephard et al. (2019) for more details). The CrIS-FPR retrieval is a physical retrieval based on the optimal estimation method (Rodgers, 2000) that uses the fast Optimal Spectral Sampling(OSS) OSS-CrIS (Moncet et al., 2008) forward model to mini-

10 mize the residual between the measured and simulated spectra. The previous version of the CrIS-NH$_3$ product (Version 1.3) has been compared with ground-based FTIR observations, and showed good performance (Dammers et al., 2017b) with correlations of r$\sim$0.8 and a slope of 1.02. The validation study indicated that for total columns greater than $10 \cdot 10^{15}$ molecules cm$^{-2}$ CrIS-FPR values have a small $\sim$0-5% difference with a standard deviation (std) of 25 up to 50%, which is within the FTIR





and CrIS estimated retrieval uncertainties. For the total columns smaller than $10 \cdot 10^{15}$ molecules cm$^{-2}$ the difference is larger, with a positive bias of $2.4 \cdot 10^{15}$ ($\pm 5.5 \cdot 10^{15}$) molecules cm$^{-2}$, equivalent to a relative difference of around 50% ($\pm$100%). This larger relative bias can occur for observations close to the detection limit of the CrIS instrument, which conservatively is about 0.9 ppb as reported by Shephard and Cady-Pereira (2015), but under ideal conditions reaches down to $\sim$0.3 ppb (Kharol

et al., 2018) (Assuming that 1 ppb is equal to a total column of $2\pm1$x$10^{15}$ molecules cm$^{-2}$ this leads to a detection limit of $1.8\pm0.9$x$10^{15}$ molecules cm$^{-2}$). Note, Version 1.5 of the CrIS NH$_3$ retrievals only contains values that have a detectable NH$_3$ spectral signal, thus in regions where there is a significant fraction of the values below the detection limit the mean values can be biased towards values at or above the detection limit (see Shephard et al. (2019) for more details). For this study only daytime observations between Jan 2013 and December 2018 are used. Furthermore, only observations with a Quality Flag of 5

are used, the dataset was further filtered by only allowing observations that have a signal-to-noise ratio > 2 (which in practice means that all observations with the unpolluted a-priori are filtered out), a degrees-of-freedom > 0.8 and a thermal contrast above >-2K, to filter out anomalous values due to thin clouds, very cold surfaces, and observations with low information content. Figure 1 shows a 5-year mean (2013-2017) of CrIS total column concentrations on a 0.05° x 0.05° grid. The data has been oversampled (Fioletov et al., 2011) with a 0.15° radius which is comparable in size to the pixel footprints. The black circles

indicate the source locations used in this study with successful emission estimates.

### 2.1.2  IASI

The IASI instruments on-board the Metop-A and -B satellites are also in sun-synchronous orbits and pass locations twice daily with an equator crossing time at 9:30 and 21:30 local solar time with a difference of about a 45 minutes (Clerbaux et al., 2009). Both IASI instruments have an observational swath width of over 2000 km and have a pixel footprint of around 12

km diameter at nadir viewing angles. The footprints are larger to the edges of the swaths with the outermost pixels having a footprint of 20 x 39 km$^2$. The ANNI-NH$_3$-v2.2 retrieval product (here called IASI-NNv2.2 unless otherwise noted) (Van Damme et al., 2017) was released in mid-2018. This product is the most recent iteration of the IASI-NH$_3$ products and an improvement over the previous datasets (Van Damme et al., 2014a; Whitburn et al., 2016a). The IASI-NNv2.2 product uses a neural network to link the hyper-spectral range index (HRI) with a set of parameters, representing the atmospheric state, to

derive the total columns NH$_3$. The near-real time IASI-NN data products have small discontinuities in the dataset following updates on the meteorological parameters provided by EUMETSAT. The IASI-NNv2.2 product improved this by providing a reanalysis dataset (ANNI-NH$_3$-v2.2R-I, here called IASI-NNv2.2(R)) that allows us here to evaluate the emissions estimates from year-to-year. Unless otherwise noted we use the reanalysis dataset, which is available until the end of 2016. For 2017 we rely on the near-real-time dataset. As of yet there is no validation study focused specifically on the newest data product.

However, the previous data products of IASI have been validated by Dammers et al. (2016, 2017b) and were found to have a low bias of around 35% for the older IASI-LUT product (Van Damme et al., 2014a) depending on the local concentrations, with better performance for regions with high concentrations. The IASI-NNv1.0 product (Whitburn et al., 2016a) showed a better performance reducing the underestimation to $\sim$25-40% depending on the concentration range. The IASI instrument has a NH$_3$ detection limit of around 2.4 ppb as reported by Van Damme et al. (2014a), assuming that 1 ppb is equal to a total column of



$2\pm1\text{x}10^{15}$ molecules cm$^{-2}$, we find a detection limit of $4.8\pm2.4\text{x}10^{15}$). In this study, we use the FORLI-CO product (Hurtmans et al., 2012) to detect and remove the influence of overpassing biomass burning plumes in the region surrounding the emission sources of interest. The IASI-CO product has been validated with MOZAIC-IAGOS aircraft observations and inter-compared with MOPITT observations by George et al. (2015) with the results showing an overall good performance. In this study only daytime IASI-A observations between January 2008 and December 2017 are used and only daytime observations between March 2013 and December 2017 for IASI-B. The NH$_3$ and CO products are matched on a per observation basis (using the time stamp, and the longitude and latitude of the observations) and filtered for conditions with cloud cover over 25%. A 5-year mean (2013-2017) of IASI-A total column concentrations can be found in Appendix A, Fig. A1.

## 2.2 NH$_3$ emission inventories and source locations

A list of potentially observable emission sources was created through a combination of an analysis of global yearly averaged concentration fields and emission inventories. Inspection of the concentration fields revealed that a large number of these locations correlated with known industrial locations. Van Damme et al. (2014b) showed that there are discrepancies between modeled NH$_3$ and IASI NH$_3$ observations for a few of these locations. Fertilizer production and gasification plants were common in this initial search; therefore, we compiled a list of worldwide plant locations based on Internet sources (https://globalsyngas.org/resources/world-gasification-database/) and used Google Earth to determine the exact locations. To this initial set we added locations from commonly used and readily available point source emission inventories. For Europe the European Pollutant Release and Transfer Register (E-PRTR, http://prtr.ec.europa.eu/) is used, for Australia the National Pollutant Inventory (NPI, http://www.npi.gov.au/), for Canada the National Pollutant Release Inventory (NPRI, https://www.canada.ca/en/services/environment/pollution-waste-management/national-pollutant-release-inventory.html), and for the United States the National Emissions Inventory (NEI, https://www.epa.gov/air-emissions-inventories/national-emissions-inventory-nei). From the point source databases any locations with NH$_3$ emissions over 0.3 kt yr$^{-1}$ were selected. While this would be of an order lower than the expected lower limit of what IASI and CrIS can detect (see Table 2 in section 2.5.2), many of the point source emissions are not measured but rather based on engineering estimates that can have up to an several orders of uncertainty (Kuenen et al., 2014; Van Damme et al., 2018). Finally, any locations from the set reported by Van Damme et al. (2018) that were missing from our set have been appended to the final location list.

For a complete list of final locations see the supplementary material. The method used in this study requires a point source like emission source. Therefore, the focus of this study is on emissions sources that behave as a point source, and not regionally sized agricultural emissions or large regions with a considerable number of point sources near one another. Emission estimates will still be attempted for sources that fail the criteria, such a large number of locations in for example the Netherlands, China and India where extended regions with high NH$_3$ concentrations exist. Several locations found in the point source emission databases are within 15 km of one another. In such instances the locations are merged under a single location name with the location representing multiple sources.

The satellite emission estimates will also be compared to the commonly used Hemispheric Transport Atmospheric Pollution version 2 emission inventory (HTAPv2, edgar.jrc.ec.europa.eu/htap_v2/, Janssens-Maenhout et al. (2015)) and the individual


point source emission databases where available (e.g. the 2014 NPRI, NEI, E-PRTR and the NPI inventories). The HTAPv2 database covers the year 2010 and provides emissions with a high spatial resolution of $0.1° \times 0.1°$. The database consists of a combination of gridded regional emission inventories such as the MICS for Asia, EPA for US and Canada and the TNO-MACCII database for Europe. These high spatial resolution inventories were complemented by the EDGARv4.3 database (Crippa et al., 2016) to gap fill missing emissions for countries not covered by the high-resolution databases.

## 2.3 Meteorology

In this study a wind rotation approach is used (Pommier et al., 2013; Fioletov et al., 2015; Clarisse et al., 2019), which requires the wind fields for each satellite observation, in order to align the observations in an upwind/ downwind coordinate system and thus map out the emission plume. Here, we use the wind fields (U,V) from ECMWF interim dataset (http://dataportal. ecmwf.int/data/d/interim_full_daily, Dee et al. (2011)) at a resolution of $0.75° \times 0.75°$ resolution ($\sim$40 x 40 km$^2$) with a 6-hour temporal resolution. Most of the observed $NH_3$ is located in the lower boundary layer (Dammers et al., 2017a; Zhang et al., 2018b; Tevlin et al., 2017). To represent the transport of $NH_3$ as well as possible we choose to use an average of the wind fields for the first kilometer of the lower troposphere. We use the 1000-900 hPa layers for locations at sea level altitude, and use 100 hPa of pressure layers for locations at higher altitudes. The first kilometer represents the intermediate case between the potentially lower boundary layers in the morning (IASI overpass, 9:30) and the higher boundary layer heights in the late afternoon (CrIS overpass, 13:30). The wind fields are then interpolated to the position of each individual observation in both space and time. The higher resolution ERA5 (Copernicus Climate Change Service (C3S)) U,V fields were also tested for this application, however, we found highly variable winds near terrain features, such as hills and coastlines, and therefore not representative of larger scale transport.

## 2.4 Data Criteria

The satellite observations measure in the infrared portion of the radiation source, which is emitted from the Earth. Thus, under colder conditions the thermal signal is decreased while the instrument noise remains the same, which reduces the overall signal-to-noise ratio (SNR) and sensitivity. Therefore, in this study we used only observations during warmer conditions between the 1st of April and the 30th of September in order to optimize conditions to which the satellites are more sensitive for the Northern Hemisphere. Similarly, only observations made between the 1st of October and the 31st of March are used for the Southern hemisphere. Besides the individual satellite product quality filters, we also correct for the influence of local events with anomalous concentrations, such as manure spreading and fire emissions, by applying a combination of two standard deviation filters. For each of the individual $NH_3$ emission sources a sub-region is defined that spans $4° \times 6°$ using the complete ten year dataset of the IASI-A instrument. Each set is then analyzed for $NH_3$ and CO anomalies, defined as the mean of all observations $\pm 5\sigma$ and $\pm 3\sigma$. The corresponding days with unusual $NH_3$ and CO concentrations and $\pm 1$ days are filtered from the dataset. Figure 2 shows an example of the data selection for the location of Togliatti in Russia. The total column time series clearly shows the anomalously high total column concentrations during the Russian Fires in 2010 (R'Honi et al., 2013), which are omitted from our data set using the previously explained steps. Finally, the list of omitted days found in for IASI-A is used



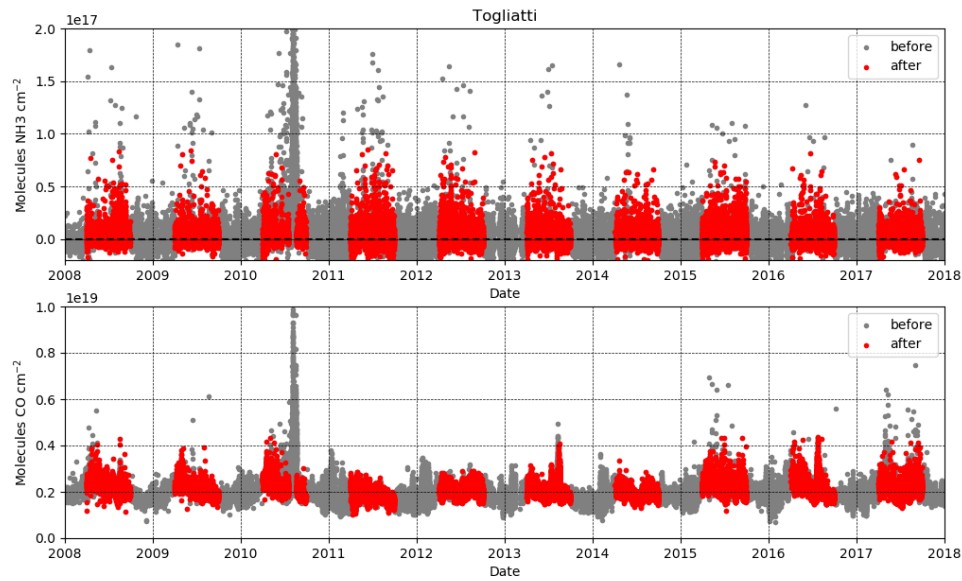

**Figure 2.** Data selection example for Togliatti, Russia. Top plot shows the IASI-A NH$_3$ dataset "before" observations are filtered out (grey) and the observations kept "after" (red) the selection. The bottom plot shows the corresponding IASI-A CO observations.

to filter the same observation days from both the IASI-B and CrIS datasets to create consistency between the datasets. Note, there is currently no global CrIS-CO dataset available to independently apply the same steps to CrIS-NH$_3$.

## 2.5 Method and uncertainties

### 2.5.1 Wind rotation and fitting algorithm

5   Previous NH$_3$ emission studies used a simple mass balance in a pre-defined box with a specified lifetime (Jacob, 1999; Van Damme et al., 2018; Hickman et al., 2018). Several other methods have been utilized in the past to retrieve emissions of other molecules from satellite observations (Fioletov et al., 2015; Beirle et al., 2014; Fioletov et al., 2011; de Foy et al., 2015) adding more complexity over time such as wind rotation to account for basic advection and diffusion. Here, we follow the wind rotation approach, and apply the Exponentially-Modified-Gaussian (EMG) method, to estimate the lifetime, the $\sigma$, and the emission of

10  the point source. The EMG method (see Appendix B for a detailed EMG description) has been shown to be accurate by de Foy et al. (2014) and has previously been used to estimate SO$_2$ emissions (Fioletov et al., 2015). It should be noted that, while accurate, the method is sensitive to wind direction and wind speed uncertainties and works best for locations with unobstructed homogeneous terrain. It tends to (overestimate) underestimate lifetime in cases of plume rotation by (overestimating) underestimating wind speeds, with a compensating (decrease) increase in the fitted width parameter. Therefore, we added the altitude





and altitude standard deviation above 300m for the surrounding 1°x1° to the location list in the supplementary material. For the altitude information we used the GLOBE product (https://www.ngdc.noaa.gov/mgg/topo/globe.html, Hastings and Dunbar (1999). Keep in mind that for some of these locations it is still possible to perform an estimate, however the results may be less reliable. Observations with wind speeds below 0.5 and above 44.5 km h$^{-1}$ are filtered from the set to remove abnormally high

concentrations during stagnant, and low concentration in highly disturbed conditions. To improve signal-to-noise, individual CrIS and IASI observations are binned in space (3 km x 4.5 km [crosswind, downwind]) and by wind speed (in steps of 2 km hr$^{-1}$ ranging from 0.5 to 44.5 km hr$^{-1}$). For each grid cell we take all observations within a radius of 15 km around the center of the grid cell, with each observation weighted by a Gaussian weighting function that takes into account the distance from the observation to the grid cell center. We can then fit an EMG function describing the concentrations near the source to the newly

found distribution.

An example demonstrating the procedure and resulting computed emissions for the ZMU fertilizer plant in Kirovo-Chepetsk, Russia, is shown in Figs. 3 and 4, using CrIS 2013-2017 data. The weighted means of the original dataset and rotated dataset are shown in Fig.4 (a) and (b), respectively. The fitted dataset in longitude and latitude domain and in the upwind/downwind

domain are shown in Fig. 4 (c) and (d), with the location of the Kirovo-Chepetsk point source highlighted as a cyan dot. The dashed black box prescribes the area used to fit the plume. This is an example of a weaker source with a well-known source location for which the rotation still shows a roughly defined plume shape. Figure 4(a) shows the corresponding mean cross section of the binned observations in the downwind direction. Each of the lines corresponds to the mean of a different wind speed interval. It shows that the peak of the plume moves downwind and decreases in amplitude with increased wind speeds

as expected. Equation B1 is fitted to the cells that fall within the striped black square and we find a lifetime (1/$\lambda$)=3.43 hr, a $\sigma$=10.03 km, a background of 9.85 x 10$^{15}$ molecules cm$^{-2}$ and total emissions of 8.93 kt yr$^{-1}$. The uncertainty shown in the figure only includes the uncertainty of the fit. More on the uncertainty of this method can be found in section 2.5.2. Figure 4(b) shows the corresponding fit at different wind speeds, with a relatively good fit to the initial distribution. The retrieved parameters can be used to reconstruct the plume-shape and initial distribution, shown Fig. 3 (c) and (d). As the background

parameter is a single value the reconstruction will be missing some of the smaller concentration peaks surrounding the source location seen in the top left and right plots.

### 2.5.2 Method uncertainty and detection limit

In order to estimate the total uncertainty we describe and add together (in quadrature) the individual sources of error and uncertainty, as listed in Table 1.

### Satellite total columns

The dominant cause of uncertainty is the uncertainty in the total column of the satellite product itself. The emission estimates are directly related to the total mass observed by the satellite, and therefore, the uncertainty in the total columns can be directly added to the total. Random errors would average out from the large number of observations used in the fit, however, the system-

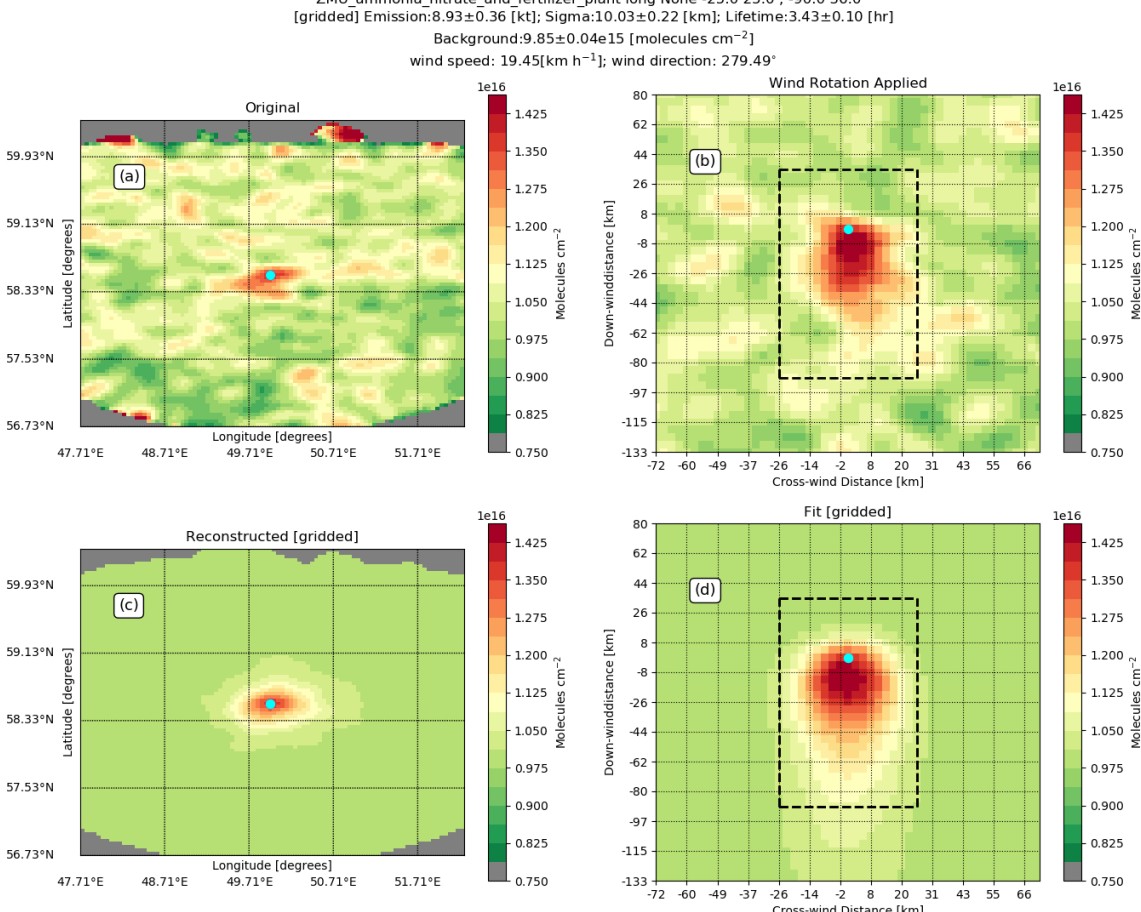

**Figure 3.** Fit example for the fertilizer plants in the ZMU fertilizer plant in Kirovo-Chepetsk, Russia. The top left plot shows the original CrIS NH$_3$ total columns [2013-2017] gridded at a 0.05° x 0.04° [lon, lat] resolution with a 15 km oversampling length to represent the footprint. The top right plot shows the same observations after applying the wind rotation algorithm and gridding the observations to a 3 km x 4.5 km grid. The bottom right plot shows the fit to the observations and the bottom left plot the reconstruction of the initial plot using the parameters found in the fit. The cyan dot indicates the source location.

atic biases do not. Dammers et al. (2017b) found a 25-40% negative bias for IASI compared to FTIR observations depending on the total column concentration. Here we will assume the lower limit of 25%, as most of the enhancements analyzed in this study have mean total column densities corresponding to the lower range of the bias. Another reason to assume the lower range is that the IASI product used here is an update over the product used in the validation, with much improved performance (Van 5 Damme et al. (2017), Figs. 1,2,3). The CrIS-NH$_3$ observations have a small bias with respect to in situ data (Dammers et al.,

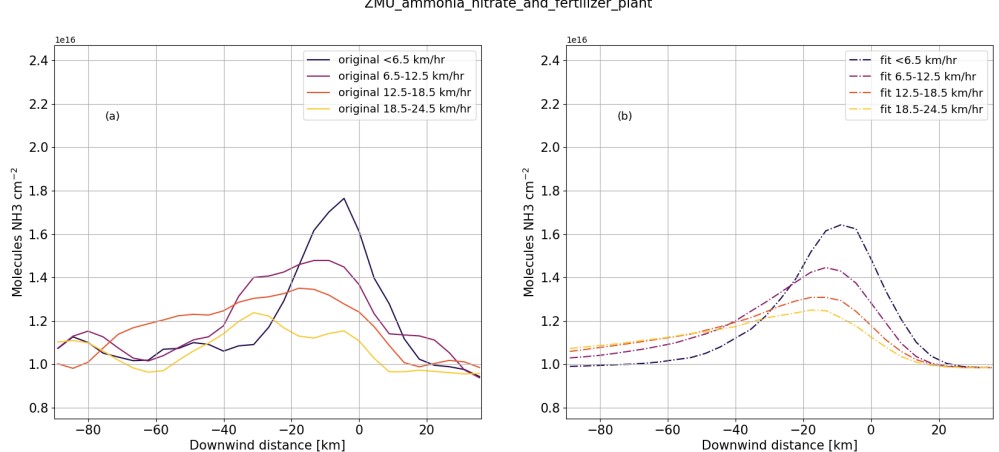

**Figure 4.** Mean total column cross-sections of the original and reconstructed plume shape for the ZMU fertilizer plant in Kirovo-Chepetsk, Russia. The left plot shows the original CrIS NH$_3$ total column distribution [2013-2017] as a function of wind speed. The cross-section is a cross-sectional average of the striped black square shown in Fig. 3. The right plot shows the mean cross-section of the reconstructed plume.

2017b) on the order of 0-5%; however, to account for retrieval errors we use an uncertainty of 16.5% following the recent emission study by Adams et al. (2019).

### Diurnal and seasonal emission cycles

The observations used here are only representative of the emissions during and before the overpass. A strong diurnal cycle in the emissions is a potential cause of uncertainty. Van Damme et al. (2018) used a model run to estimate the offset between the daily mean total column and the overpass time of IASI-A and found an uncertainty of 4.0±8.0%. Using simulations for 2014 for the same model we an uncertainty of 6.3±7.1% for the IASI-A overpass time and 8.0±10.5% for the CrIS overpass, with the difference following from the selection of only land based grid cells and a different simulation year (2014 vs 2011). The

model used in their study attributes a single diurnal emission cycle to all species emitted by the industrial sources emission class. However, for most industrial sources of NH$_3$, it is highly uncertain what the actual emission cycle is, as in most cases it has not been measured. As for agricultural emissions, the hourly emissions can vary strongly with temperature and soil conditions (Sommer et al., 1991). Furthermore, the emissions will vary a great deal by month, depending again on the temperature and the season in which manure and/or artificial fertilizer is spread over crop and grasslands. As our estimates are only based

on satellite measurements done within most of the growing season (Northern Hemisphere) the agricultural estimates can potentially be biased high by 50% if all emissions take place between the 1st of April and the 30th of September. Furthermore, our method weighs towards the months with more observations. Van Damme et al. (2018) found that IASI will get emission estimates that are 4±7% lower when first binning to monthly averaged grids.





**Fitting algorithm**

The fitting algorithm is another cause of uncertainty. In this study we first will first do free fits, which include a fit of the life-time and $\sigma$ as well as the emitted amounts and the background (Section 3.1). Secondly, using a mean lifetime and $\sigma$, calculated from the results of the free fits, we will recalculate the estimates with fixed fits, where the lifetime and the $\sigma$ are fixed (Section 3.2). The uncertainty of the fits is dependent on the amount of information available and the quality of the total column field: in this free fit case (Fig. 5) we find a mean fitting uncertainty (standard deviation of the fit) of 6.7-8.6% with the lower range for the IASI instruments. For the fit with fixed parameters, we find a reduced uncertainty of 3.1-4.1% with the lower range for the IASI instruments. However, any deviations due to the fixed $\sigma$ and lifetime parameters need to be considered. To calculate the effect, we perturb the parameters, by adding a random value taken from a gaussian random distribution, with a standard deviation found in Table 3, and estimate the emissions. For both the lifetime and $\sigma$ value (while keeping the other parameter constant) this process is repeated a thousand times for three different locations representing a range of emission enhancements (Donaldsonville, USA, Lethbridge, Canada, and Togliatti Russia). Taking the standard deviation of the set of estimated emissions for each location and dividing it over the mean we find a calculated uncertainty for the lifetime and $\sigma$ of $\pm29.0\%$ and $\pm40.4\%$ for CrIS and $\pm40.1\%$ and $\pm39.2\%$ for IASI.

**Meteorology**

Finally, the uncertainty in the meteorology has a direct effect on our error estimates through Eq. B3. Assuming an uncertainty of $\pm1$ m s$^{-1}$ in the wind fields, we calculate the uncertainty by using a similar method as for the lifetime and $\sigma$. A random value, taken from a Gaussian random distribution with a standard deviation of $\pm1$ m s$^{-1}$, is added to both the u and v wind field parameters that were matched to each satellite observation in the set. The emission is then estimated using the perturbed wind fields and a fixed lifetime and $\sigma$ parameters of 2.5 hours and 15 km. This process is then again repeated a thousand times for the three different locations (Donaldsonville, USA, Lethbridge, Canada, and Togliatti, Russia). Taking the standard deviation of the estimated emissions for each location and dividing it over the mean we find a calculated uncertainty of 2.6% for IASI and 3.1% for CrIS.

**Detection limit**

The EMG approach can be used to estimate the emissions for a large range of sources, but is limited by a lower detection limit that is directly related to the instrumental detection limit of the satellites. Using Eq. B1 the limit can be directly estimated over a range of conditions, and summarized in Table 2 for the IASI and CrIS instruments. The CrIS instrument has a detector with about four times lower spectral noise than IASI that results in the lower limit of 0.9 ppb under high thermal contrast (TC) atmospheric conditions reported by Shephard and Cady-Pereira (2015) compared to the limit of 2.4 ppb under high TC conditions reported by Van Damme et al. (2014a). Kharol et al. (2018) reports that CrIS is capable of measuring concentrations down to even 0.2-0.3 ppb under very favorable conditions, but here a more conservative limit of 0.9 ppb is assumed to cover most of the conditions found in the observations. Assuming that 1 ppb is equal to a total column $2\pm1\mathrm{x}10^{15}$ molecules cm$^{-2}$ we find lower limits of $4.8\pm2.4\mathrm{x}10^{15}$ and $1.8\pm0.9\mathrm{x}10^{15}$ molecules cm$^{-2}$. The ratio is highly variable and dependent on the shape





**Table 1.** Summary of factors of uncertainty in the final satellite emission estimates for IASI-A, -B and CrIS.

| Source of uncertainty | IASI-A (%) | IASI-B (%) | CrIS (%) |
|---|---|---|---|
| **Total column from satellite** | ±25.0 | ±25.0 | ±16.5 |
| **Wind speed and direction** | ±2.6 | ±2.6 | ±3.1 |
| **Diurnal variability†** | 6.3 | 6.3 | -8.0 |
| **Seasonal emissions‡** | -50.0 | -50.0 | -50.0 |
| Lifetime | ±40.1 | ±40.1 | ±29.0 |
| $\sigma$ | ±39.2 | ±39.2 | ±40.4 |
| *Fit (free)* | ±6.7 | ±6.8 | ±8.6 |
| *Fit (fixed)* | ±3.1 | ±3.2 | ±4.1 |
| **Method total (free)[a]** | ±6.7 | ± 6.8 | ±8.6 |
| **Method total (fixed)[b]** | ±56.2 | ±56.2 | ±49.9 |
| **Total uncertainty (free)[c]** | (-26.0, 26.8) | (-26.0, 26.8) | (-20.5, 18.9) |
| **Total uncertainty (fixed)[d]** | (-61.5, 61.9) | (-61.5, 61.9) | (-53.3, 52.6) |
| **Total uncertainty agricultural (free)[e]** | (-56.4, 26.8) | (-56.4, 26.8) | (-54.0, 18.9) |
| **Total uncertainty agricultural (fixed)[f]** | (-79.3, 61.9) | (-79.3, 61.9) | (-73.0, 52.6) |

† Diurnal cycle via modelling estimate, does not include potential effect of biases in diurnal emission profiles.

‡ For agricultural areas most emissions are in the spring and summer hence these are influenced by the fact that our estimate is for those months only.

[a] Sum of the uncertainties from Fit (free).

[b] Sum of the uncertainties from Fit (fixed), Lifetime and $\sigma$.

[c] Sum of the uncertainties from Method (free), Total Column, Wind speed and Diurnal variability.

[d] Sum of the uncertainties from Method (fixed), Total Column, Wind speed and Diurnal variability.

[e] Sum of the uncertainties from Method (free), Total Column, Wind speed, Diurnal variability and Seasonality.

[f] Sum of the uncertainties from Method (fixed), Total Column, Wind speed, Diurnal variability and Seasonality.

of the atmospheric profile, therefore we add a ±50% range. Furthermore, a $\sigma$, and lifetime of 15 km and 2.5 hours respectively and three average wind speeds of 5, 10 and 20 km hr$^{-1}$ are assumed, which are representative values for most of the sources as shown in section 3. The lowest emission detection limits are naturally found for the lowest mean wind speeds with a detection limit of 9.5±4.7 molecules cm$^{-2}$ for IASI and 3.5±1.8 for CrIS molecules cm$^{-2}$. These estimates are a conservative lower limit as plume shapes are not as well defined making emission estimates less precise. The limits for the 10 and 20 km hr$^{-1}$ cases should be considered as more representative of conditions commonly occurring at most locations.





**Table 2.** Estimated emission detection limit

| Satellite instrument | Observation detection limit | | Reference | Emission detection limit [kt yr$^{-1}$] | | |
| | ppb | molecules cm$^{-2}$ | | 5 km hr$^{-1}$ | 10 km hr$^{-1}$ | 20 km hr$^{-1}$ |
| --- | --- | --- | --- | --- | --- | --- |
| IASI | 2.4 (4.3) | *4.8±2.4x10$^{15}$ | Van Damme et al. (2014a) | 9.5±4.7 | 13.6±6.8 | 22.4±11.2 |
| CrIS | 0.9 (0.3) | *1.8±0.9x10$^{15}$ | Shephard and Cady-Pereira (2015) | 3.5±1.8 | 5.1±2.6 | 8.4±4.2 |

*assuming a conversion of 1 ppb $\sim 2\pm1$x10$^{15}$ molecules cm$^{-2}$

## 3 Results

### 3.1 Lifetime and emission fits

The emission algorithm is applied to each of the source locations obtained as described in section 2.2 and listed in supplementary material for the 5-year interval (2013-2017) of IASI-B and CrIS and the 5- and 10-year (2013 & 2008 to 2018) intervals of IASI-A data. For quality assurance of the estimated emissions, we filter out low quality fits with $r < 0.5$ and an upwind-downwind signal-to-noise ratio SNR<2. (McLinden et al., 2016), defined as;

$$SNR = \frac{\bar{C}_d - \bar{C}_u}{\frac{\sigma_d}{\sqrt{N_d}} + \frac{\sigma_u}{\sqrt{N_u}}} \qquad (1)$$

$\bar{C}_d$ and $\bar{C}_u$ are the mean down- and up-wind total columns, $\sigma_d$ and $\sigma_u$ are the standard deviations, and $N_d$ and $N_u$ are the number of down- and up-wind values. Here we use the SNR filter to remove locations with small enhancements and high regional backgrounds to provide results from single sources with large and sharply defined enhancements (McLinden et al., 2016). Furthermore, fits with lifetimes below 1 hour and above 7 hours, and $\sigma$ below 5 km and above 30 km, were filtered out as well. All four values represent the edges of their respective distributions and any value past these edges resulted in bad fits.

Figure 5 shows the results for all successful fits for CrIS data over the 5-year (2013-2017) interval. The resulting set has a mean lifetime of 2.35±1.16 hr and a mean $\sigma$ of 16.42±5.13 km.

**Table 3.** Mean lifetime and $\sigma$ for all successful fits with a correlation > 0.5 and a SNR > 2 for the 5-year IASI-A, IASI-B and CrIS and 10-year IASI-A datasets

| Satellite | Period | N | Mean lifetime [hr] | Mean $\sigma$ [km] |
| --- | --- | --- | --- | --- |
| CrIS | 2013-2017 | 83 | 2.35±1.16 | 16.42±5.13 |
| IASI-A | 2008-2017 | 87 | 2.03±1.05 | 16.24±5.78 |
| IASI-A | 2013-2017 | 62 | 2.22±1.25 | 16.08±5.39 |
| IASI-B | 2013-2017 | 58 | 2.09±0.84 | 15.67±5.69 |

Table 3 and Figs. C1, C2 and C3 in Appendix C show the corresponding results for IASI-B, and the five- and ten-year interval IASI-A sets; all show similar results, which might be expected for effective sigma given the similar footprint sizes of





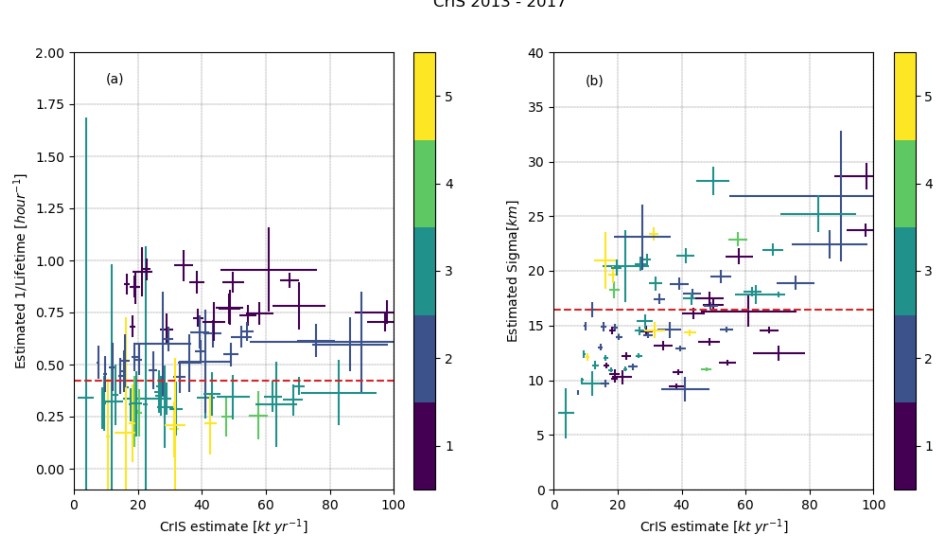

**Figure 5.** All successful fits for the 5-year (2013-2017) CrIS dataset with a correlation > 0.5 and a SNR > 2. The left plot (a) shows the fitted emissions versus the $\lambda$ parameter and the right plot (b) the fitted emissions vs the fitted $\sigma$ with the horizontal and vertical error bars showing the uncertainties of the fit. The color of the bars indicates the fitted lifetime (in hours) rounded to the nearest integer. The horizontal striped red lines show the average of the $1/\lambda$ and $\sigma$ parameters.

the instruments. For all sets we find a lifetime between 2.03 - 2.32 hours with a standard deviation ∼1 hour. This is shorter than most of the measured and modelled estimates reported in literature, ranging from a few hours up to 2 days. Most of the longer lifetimes were determined by analyzing short and long range transport of fire plumes, emitted near the surface or injected at higher altitudes (Yokelson et al., 2009; R'Honi et al., 2013; Whitburn et al., 2015; Lutsch et al., 2016; Adams et al., 2019),

which are of limited applicability here, as the high concentrations of other species found in a plume that gets injected into the free troposphere, are not very comparable to normal surface conditions. The only other study reporting satellite estimates of $NH_3$ for agricultural and industrial sources, Van Damme et al. (2018), used a lifetime of 12 hours quoting the limited data available in literature. Van Damme et al. (2018) also calculated emissions for lifetimes of 1 and 48 hour as an upper and lower estimates which will be of more use for later discussion. Nonetheless, the decay of the $NH_3$ plume over a distance ∼80 km as

seen in Fig. 4 (which is typical) suggests an effective lifetime on the order of 2-4 hours. The mean sigma of ∼15 km is in line with earlier estimates for $SO_2$ (Fioletov et al., 2015), with the sigma mostly varying with spatial extent of the source, differences in diffusion for each location, and the influence of nearby sources. The slightly larger value for CrIS is consistent with its slightly coarser spatial resolution. The wide distribution around the mean can potentially also follow from under-constrained fits as well as the large spatial extent of some of emission sources. As seen in Figs. 5(b) (and Figs. C1, C2 and C3 in Appendix

C) the $\sigma$ parameter shows a positive relation with the total estimated emission. The largest emission estimates are for agricultural hotspots, which in some cases are poorly described as single point sources and any result for such examples should be





treated with caution.

## 3.2 Constrained fits and comparison

As indicative by the limited number of points shown in Figs. 5 (and C1, C2, and C3 in appendix C), only a few locations have

5    plume shapes with large enough total column enhancements for estimating the lifetime $1/\lambda$, the spread $\sigma$, background B, and emissions $a$, which require a non-linear, and hence less stable fit. To increase the number of converging fits the lifetime and $\sigma$ parameters are fixed to 2.5 hours and 15 km, similar to the weighted means found in Table 3, thus requiring only a linear fit. The algorithm was then reapplied to all locations in order to determine emission estimates for those with weaker enhancements in which the original fits failed due to a lack of independent information to fit these additional parameters. The results of those

10    fits are quality controlled as before (see section 3.1) but now with a weaker correlation filter, removing only observations with a fit correlation of r<0.30. All results are merged into the location list and can be found in the to supplementary material.

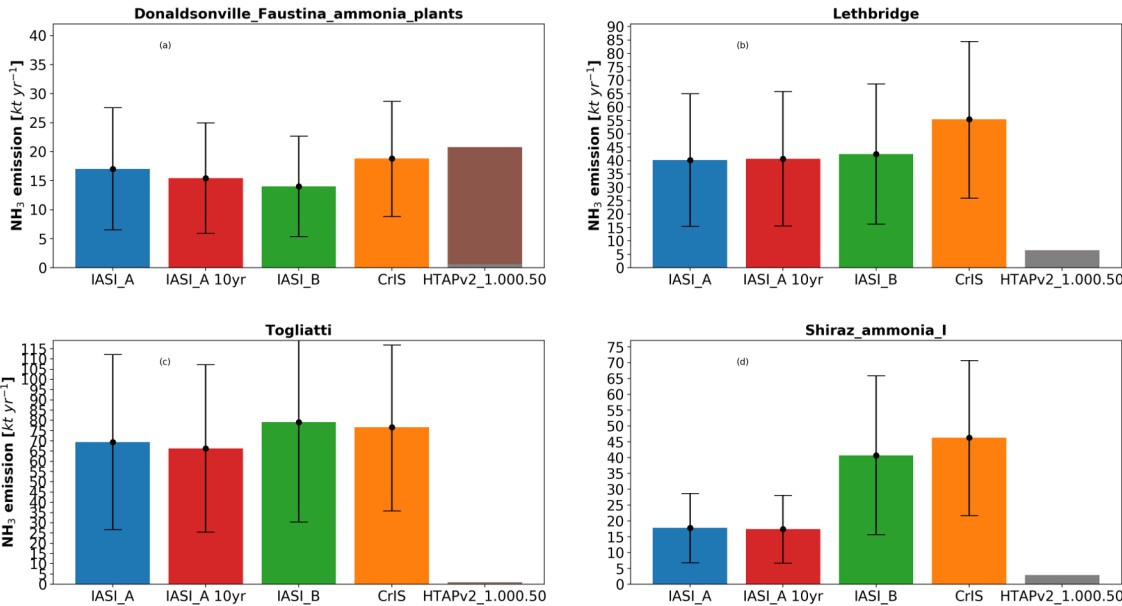

**Figure 6.** Comparison of the IASI-A (5yr [blue] and 10yr[red]), IASI-B[green] and CrIS[orange] emissions estimated using fixed parameters for lifetime and $\sigma$, and the corresponding total emission found in the HTAPv2 inventory for a square $1.00° \times 0.50°$ [longitude,latitude] surrounding the location. The grey part of the bar indicates the agricultural emission total while the brown section represents all other sources.

Figure 6 demonstrates a few examples of emissions from the fit results compared to the emissions calculated from the HTAPv2 emission inventory and integrated over $1.00° \times 0.50°$. Figure 6 (a) shows results for the $NH_3$ production plants in Donaldsonville and Faustina in the United States, with all four satellite estimates showing comparable results. This is one of





the few examples that compares well with the current inventory estimates, as can be seen from the other three examples. Figure 6 (b) shows the results for Lethbridge, Canada, where a large number of concentrated animal feeding operations (CAFOs) can be found. The estimates compare poorly with the estimate calculated from HTAPv2. The difference can be explained by a few factors: first, as only the observations between the 1st of April and the 30th of September are used, the estimate is only

representative of the spring and summertime emissions. Outside of the spring and summer period, emissions of CAFO's are expected to be lower due to slower volatilization rates (Gyldenkærne et al., 2005); second, differences in agricultural practices, with most fertilizer being spread in early spring (Sheppard et al., 2010) which can occur early or late depending on the year. If most emission occur during the the six spring and summer months we would need to halve the satellite estimates, which would bring the estimates closer. Figures 6 (c) and (d) show the estimates for the $NH_3$ plants in Togliatti, Russia, and near Shiraz,

Iran respectively. In three of these four cases the satellite emission estimates do not agree with the HTAPv2 inventory, with the inventory showing much lower total emissions. Most of the industrial emissions are computed using empirical estimates and are not measured (E-PRTR database), therefore large differences can be expected. The fits for the $NH_3$ production plants near Shiraz are a good example of the limited applicability of this method in regions with large elevation changes nearby and only a single outflow direction. While on the first look the fits are reasonable (see appendix Figs. D1,D2 and D3) the overall wind

speed is very low and the plume shape is not well defined. This limitation is more visible for IASI-A than for IASI-B and CrIS which can be due to small deviations in the concentration field.

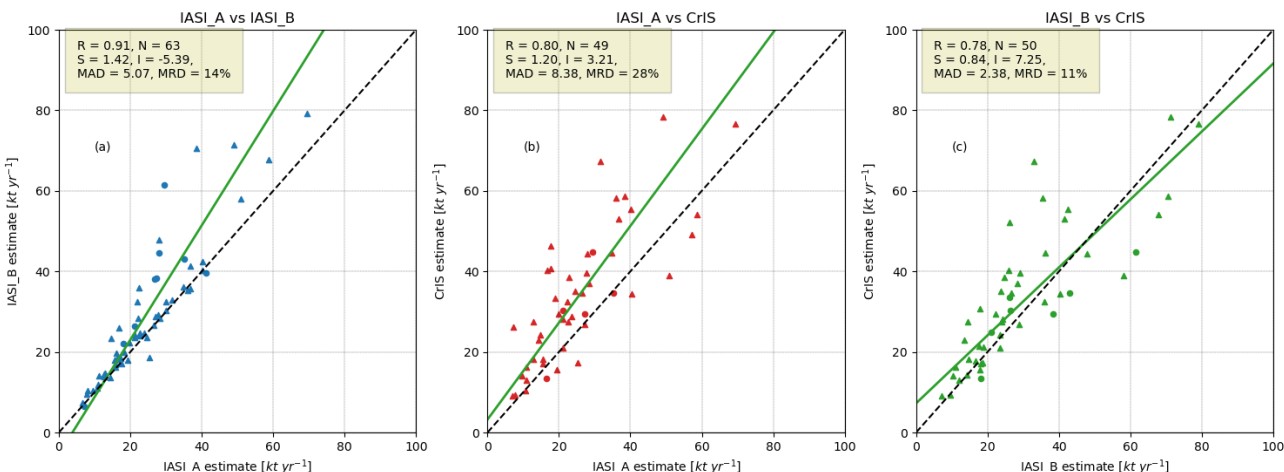

**Figure 7.** IASI-A vs IASI-B and IASI-A vs CrIS for locations with r>0.5 with a fixed sigma of 15 km and lifetime of 2.5 hours. 2013-2017. R indicates the correlation between the sets, N the number of locations, S and I the resulting Slope and Intercept of the RMA regression and MAD and MRD the mean absolute difference (y-x) and the mean relative difference (y-x/(0.5y+0.5x)). Triangles indicate locations with agricultural emissions equal to and above the sum of all other classes in the HTAPv2 inventory.

The emission estimates from the fits with fixed lifetime and spread over the locations with r>0.5 from all three satellites are in good agreement (Figure 7). IASI-A and -B compare very well with, a tight distribution around the x:y line and a few





higher estimates for IASI-B biasing the fit high. The positive bias between the IASI instruments and CrIS is consistent with the different overpass times. Compared to the IASI 9:30 overpass the top of the boundary layer will be higher at the CrIS 13:30 overpass. Especially during days with low thermal contrast in the morning IASI might not observe the lowest layer of the troposphere and miss a fraction of the atmospheric columns, leading to an underestimation of the emissions. Emissions from

agricultural sources are also expected to be higher in the afternoon because of higher temperatures, which increase volatilization rates. Lastly, while there is a bias between the CrIS and IASI-A and -B total columns, an offset between the products does not have too much of an impact on emissions, as this should be caught by the background parameter. Any ratio between the products, however, can create a ratio in the emission estimates. Dammers et al. (2017b) validated the CrIS and IASI(-A) $NH_3$ products. Compared to the FTIR-$NH_3$ product the IASI product was biased low for lower total columns while CrIS biases

high. For increasing total column densities the bias in the CrIS columns disappears while IASI shows a smaller and smaller negative bias. Coarsely transformed to an IASI to CrIS ratio it can be expected that the CrIS estimates will be higher than IASI as IASI likely underestimates the total columns and thus the emission total. As noted earlier, the validation by Dammers et al. (2017b) was for the previous versions of both the CrIS and IASI retrievals. The IASI retrieval has evolved, with the new retrieval showing much improved performance (Van Damme et al., 2017).

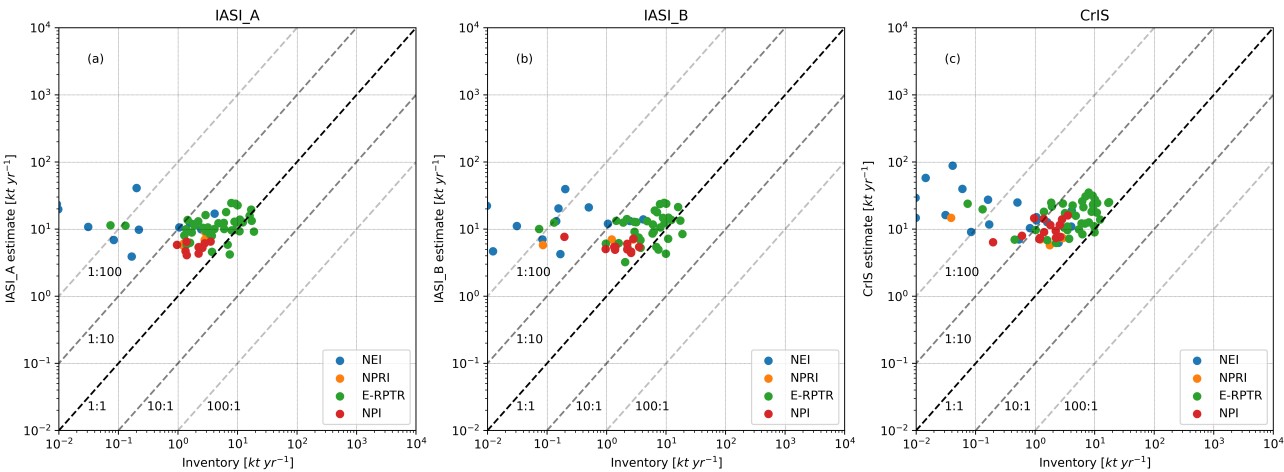

**Figure 8.** IASI-A, IASI-B and CrIS emission estimates vs the regional point source emission inventories. Point source values are the sum of all emissions in a square $0.50°$ x $0.25°$ [longitude, latitude] surrounding each source location.

Figure 8 shows a comparison of emissions between (fixed) fits to the satellite data and the point emission databases of the United States (NEI), Canada (NPI), Europe (E-PRTR) and Australia (NPI), for point sources in the emission databases. Overall, the point source inventories compare poorly to the fitted emissions. Only the E-PRTR inventory has a majority of locations for which the fitted totals are within an order of magnitude of the totals from the inventory. While an uncertainty of 100-300% is to be expected following Kuenen et al. (2014), some estimates are multiple orders larger than reported emissions. The satellite

emission fits compare slightly better to the HTAPv2 database (Fig. 9 and 10); here we summed the satellite emission fits to all





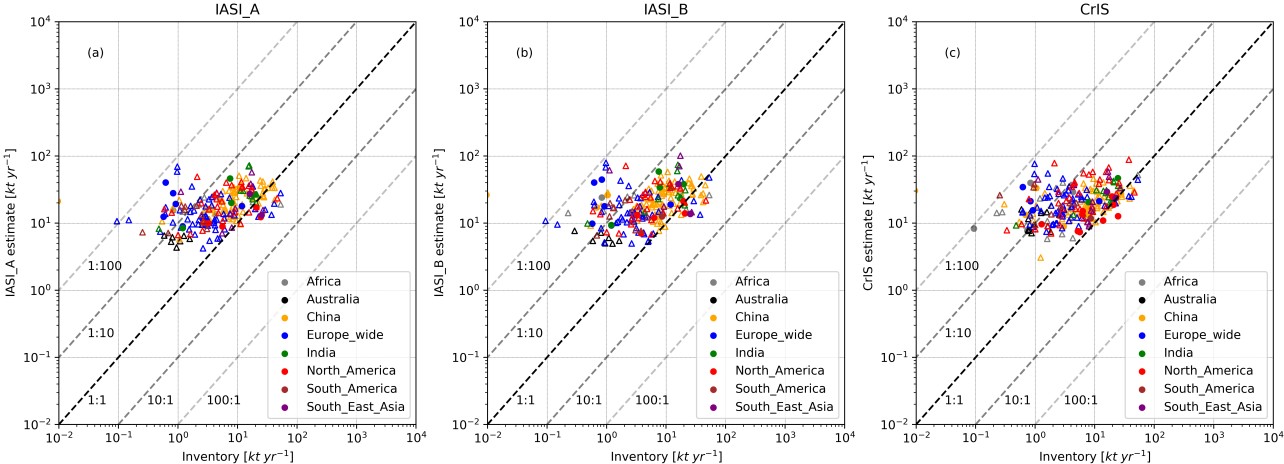

**Figure 9.** HTAPv2 emission database vs satellite estimates. HTAPv2 emissions are available globally and the sum of all emissions in a square $1.00° \times 0.50°$ [longitude, latitude] surrounding each source location. Open triangles indicate locations with agricultural emissions equal to and above the sum of all other classes in the inventory.

HTAPv2 emissions in a $1.00° \times 0.50°$ box surrounding the source location. However, here we also find that in 55 (IASI-A:33, IASI-B:39) cases the inventories emissions are either missing or underestimated by more than an order of magnitude. 72 (IASI-A:71, IASI-B:64) locations are within a factor 2 of the CrIS emission estimates and in most cases represent agricultural regions (illustrated by the triangles). These results, using a more sophisticated method to estimate the satellite-based emissions, confirm the underestimation of the current emission inventory reported for EDGAR v4.3.1 by Van Damme et al. (2018). A comparison between the results of this study and those obtained using a box model approach is provided in Fig. E1 in Appendix E.

For the exact dimensions of the individual regions mentioned here see Table F1 and Fig. F1 in Appendix F. Table 4 shows the total emissions estimated with this studies compared to the total emissions in the HTAPv2 inventory. We find that our method estimates that 3537 kt yr$^{-1}$ is missing in the inventory for the locations used in this study, which converts to a factor $\sim 2.5$ between satellite estimated emissions and the current HTAPv2 emissions. Based upon the total NH$_3$ emissions in HTAPv2 the locations used in this study account for around 5% of the total yearly global emissions.



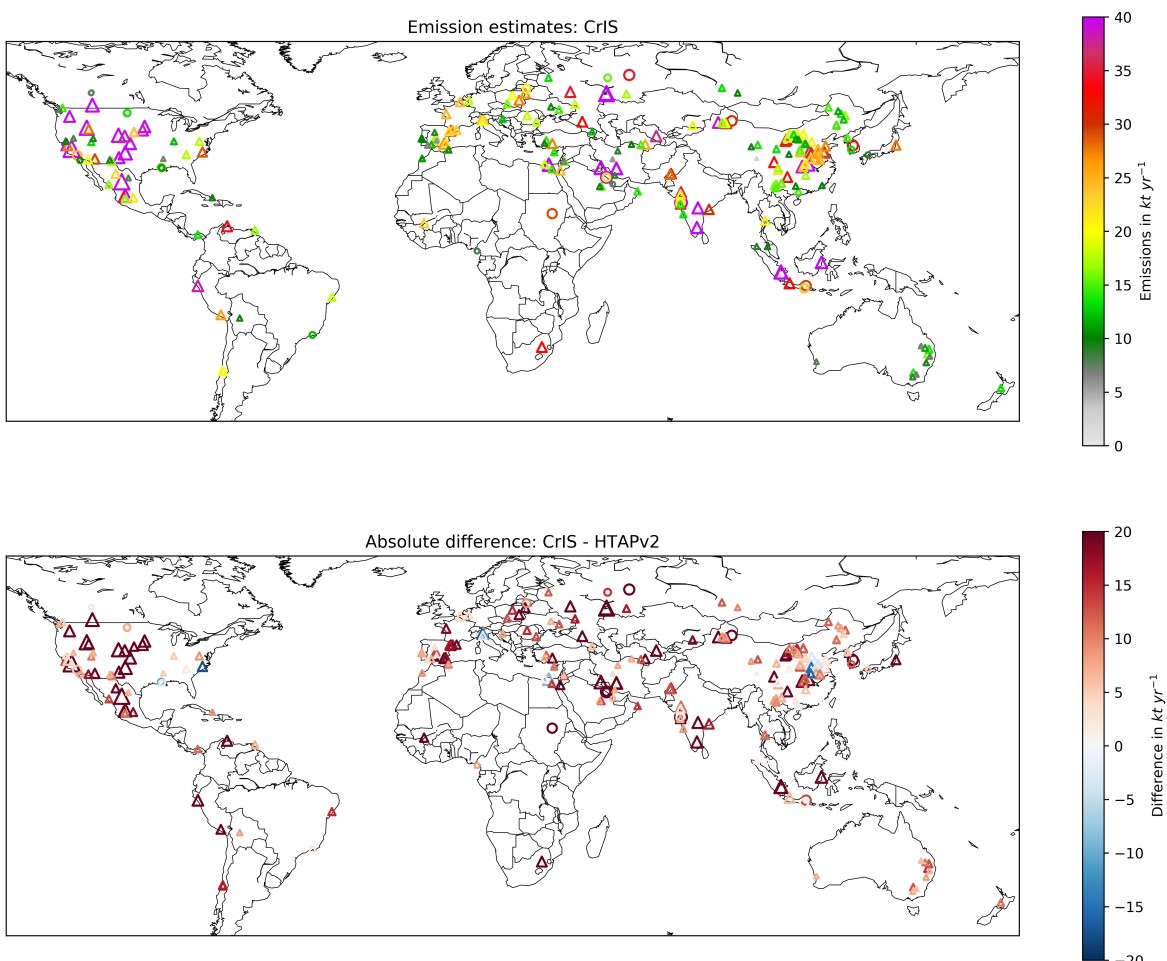

**Figure 10.** Global overview of CrIS Emission estimates compared to the HTAPv2 inventory for locations with r>0.3, SNR>2., a fixed sigma of 15 km and a lifetime of 2.5 hours. The top plot shows the CrIS emission estimates and the bottom plot the difference between the CrIS estimates and the corresponding emissions found in the HTAPv2 inventory for a square 1.00° x 0.50° [longitude, latitude] surrounding the locations. Open triangles indicate locations with agricultural emissions equal to and above the sum of all other classes in the inventory.





**Table 4.** Summary of the CrIS, IASI-A and IASI-B emission estimates per region versus HTAPv2 inventory entries for all successful fits with a correlation > 0.3 and a SNR > 2. Each entry represents the CrIS value with the IASI-A and IASI-B values following between the brackets: CrIS (IASI-A, IASI-B).

| Region | HTAPv2 [kt yr$^{-1}$] | Satellite Estimates [kt yr$^{-1}$] | Locations # | Region total HTAPv2 [kt yr$^{-1}$] | Fraction of inventory emissions [%] |
|---|---|---|---|---|---|
| **Europe** | 373 (412, 318) | 1400 (1132, 1160) | 63 (66, 60) | 9860 | 3.8 (4.2, 3.2) |
| **North America** | 445 (291, 301) | 1295 (698, 761) | 48 (33, 34) | 5123 | 8.7 (5.7, 5.9) |
| **India** | 221 (185, 196) | 408 (434, 518) | 15 (14, 16) | 14016 | 1.6 (1.3, 1.4) |
| **China** | 761 (948, 940) | 1556 (1783, 2006) | 74 (79, 84) | 20216 | 3.8 (4.7, 4.6) |
| **Africa** | 95 (91, 104) | 320 (166, 186) | 16 (9, 9) | 4929 | 1.9 (1.8, 2.1) |
| **Australia** | 14 (10, 7) | 132 (50, 42) | 12 (8, 7) | 1160 | 1.2 (0.8, 0.6) |
| **South America** | 34 (15, 31) | 207 (116, 150) | 10 (9, 11) | 4644 | 0.7 (0.3, 0.7) |
| **South East Asia** | 142 (134, 113) | 304 (227, 317) | 11 (10, 7) | 7845 | 1.8 (1.7, 1.4) |
| **Total** | **2086 (2084, 2009)** | **5622 (4606, 5140)** | **249 (228, 228)** | **48635** | **4.3 (4.3, 4.1)** |


### 3.3 Emission time series

Instead of estimating average emissions over the entire 2008-2017 interval it is also possible to calculate yearly estimates using a running mean over three years of data. Only for the strongest sources is it was feasible to estimates using only a single year of observations, as the random noise is sufficiently reduced to produce a strong signal. Therefore, instead we use a set of three years, with the edges of the series consisting of two years. By combining a manual search through the results with information on each source location, a number of interesting emissions time series were revealed. Figure 11 shows the emission time series for a number of source locations estimated from IASI-A data, which has the longest period of record of the satellite instruments used in this study. The top plot (a) shows the emission time series of locations situated in or near conflict zones. Both the Horliwka/Gorlovka and Severodonetsk petro-chemical plants are located in the east of Ukraine in or near the front line of the War in the Donbass region. The conflict started around March 2014, which coincides with the rapid decline in emissions from 2013 onward (the three year set uses the previous year), following the closure of the fertilizer production plants (Dimitry Firtash, 2014). Another example are the agricultural regions south of Homs and east of Damascus, which have both been impacted by the Syrian Civil war. The civil war started around 2011, and the reduced emissions are especially visible for the region south of Homs, which was in the rebel/opposition controlled region at the time. Similarly the agricultural region east of Damascus had reduced emissions for this period, which reflects the state of most agricultural activities throughout Syria during the war (FAO, 2017). The middle plot (b) shows the examples of the $NH_3$ factories in Nevinnomyssk, Russia, and Zalantun (Hulunbeir), China regions, which both show increasing emissions. The increase seen for the $NH_3$ plant in Nevinnomyssk follows from the increased production as of 2010/2011, after upgrades to the plant between 2008 and 2011 (EuroChem Group, 2008). The Zalantun $NH_3$ plant was built in 2010 and started production at the end of 2011 (full production start of 2012, Fufeng Group (2012)). The bottom plot (c) shows the example of a number of locations where decreasing emissions were found connected to reduced production. The Salavat, Russia, Petrochemical Complex reduced production to permit upgrades (Gazprom, 2017), and is expected to increase production from 2017 onward. The $NH_3$ plant in Turnu Magurele, Romania, decreased production for years before closing down (ICIS, 2011), following cuts to subsidized gas. The same subsidy cuts affected the plant in Bacau, Romania, which was reported by Van Damme et al. (2018). The nickel refinery in Yalubu, Australia, increased production in the early 2010s following new ownership and subsequent investments (Business Review Australia, 2010). However, on several occasions, (potential) tailing pond leaks to the Great Barrier Reef were reported (ABC News/Ben Millington, 2016) and subsequently the refinery has been (temporarily) shut down since 2016 (ABC News, 2016). Finally, there is the case of the $NH_3$ plant in El Tablazo, Venezuela, where $NH_3$ emissions have decreased since 2010 which seems to be due to the lack of fuel and mechanical issues since the start of the shortages in Venezuela in 2010 (ChemStrategy, 2018). For comparison with the yearly emission estimates mentioned by Van Damme et al. (2018) (Fig.4) see Fig. E2 in Appendix E.

Annual emissions over extended regions can also be calculated. Regions are defined in table F1 in Appendix F. Figure 12 shows the annual emissions estimated from IASI-A in each region and the emissions relative to the HTAPv2 inventories base year of 2010. Only locations with both successful 5 year IASI-A fits and with successful fits for all the ten individual years





**Figure 11.** Annual variations in the IASI-A NH$_3$ emission estimates for (a) sources in conflict zones, (b) industrial sources that increased production (c) industrial sources that reduced production.

**Figure 12.** Regionally summed annual IASI-A $NH_3$ emission estimates. Plot (a) shows the total emissions per year for each region, with N the number of sources used for each sum and S the Sen's slope estimate of the ten years of data for all cases with a significant Mann-Kendall trend test. The error bars represent the uncertainty of each of the individual sources added in quadrature. Plot (b) the emissions normalized by base year 2010, with N the number of sources used for each sum and S the Sen's slope estimate of the ten years of data for all cases with a significant Mann-Kendall trend test.

were used. Note that the point sources only constitute around 5 to 10% of all emissions in most regions, which in combination with the relatively few points in our set means that at best the increases and decreases should be seen as a tentative estimate of changes in each region and not as a significant trend of all the emissions in region. The top plot (a) shows the sum of the annual





emissions of the emission sources in each region. Plot (b) shows the same time series but normalized to the emissions of 2010. The year 2010 was chosen as this is the same as the HTAPv2 base year. To each time series the Mann-Kendall test (Kendall, 1938; Mann, 1945) is applied to calculate if there is a significant trend. In cases with a significant trend we estimate the slope using Sen's slope estimate (Sen, 1968) which is insensitive to outliers. For three regions there are significant trends, which

5 are all increasing. The China region (p<0.01) shows the most significant absolute changes, with a change of about 2.6% per year. The multi-annual increase mostly follows from an increase in fertilizer production and agricultural emissions. A similar increase is seen in the study by Warner et al. (2017), who attributed it to an increase in fertilizer use in the region. For the European region we find a very small change of about 0.9% per year (p<0.05). Finally, the South American region shows the largest relative change with an increase of about 5.0 % per year (p<0.01). The increase in South America follows mostly from

10 changes in agricultural emissions which was also reported by Warner et al. (2017) and Van Damme et al. (2018).



## 4  Summary and conclusions

In this study we presented the first $NH_3$ emission estimates based on the CrIS-$NH_3$ satellite observations, where both the emissions and lifetimes of $NH_3$ are derived simultaneously for a variety of agricultural and industrial point sources. Results are consistent between the CrIS and IASI satellites with an average lifetime of 2.4±1.2 hours and a $\sigma$ of 16.4±5.1 km for

CrIS, 2.2±1.3 hours and 16.1±5.4 km for IASI-A and 2.1±0.8 and 15.7±5.7 km for IASI-B. Using an average lifetime of 2.5 hours and a $\sigma$ of 15 km we found comparable emission totals for all three satellites with correlations of r=0.91, r=0.80 and r=0.78 between IASI-A and IASI-B, IASI-A and CrIS and IASI-B and CrIS, respectively. The CrIS emissions estimates are on average higher than the emissions derived from IASI-A and –B observations, but are within the uncertainty of the estimates. The differences in the emissions can be due to the bias between the satellite products, as well as the potential influence of the

different sampling times of the satellites in combination with the strong diurnal cycles of the emissions. With our method we found 249 sources with emission levels that are detectable by the CrIS satellite. Comparison with the HTAPv2 inventory shows that there are currently 55 sources missing or underestimated by more than an order of magnitude in the inventory, and only 72 sources have emissions that differ from the HTAPv2 inventory by less than a factor of 2. For IASI-A (and IASI-B) we found similar numbers, with respectively, 228 (228) total sources detected, with 33 (39) sources missing from the inventory, and

only 71 (64) sources where the HTAPv2 emissions fall within a factor 2 of the satellite estimates. All the sources successfully estimated with CrIS combined have a total emission of 5622 kt yr$^{-1}$ which is about ∼2.5 times more than the current amount in the HTAPv2 inventory (2086 kt yr$^{-1}$) over the same locations. Applying the same method to the longer 10-year IASI-A data set for yearly estimates, we are able to observe short- and long-term variations in emissions from both large and small sources, with emissions matching changes to production and other local developments. The temporal variations are consistent

with those earlier found by Van Damme et al. (2018). The regional trends in the emissions are similar to earlier findings based on the AIRS satellite (Warner et al., 2017). The results however, may not necessarily be reflective of the entire region as only the larger more isolated sources are included in this studies satellite estimates.

The sum of the HTAPv2 emissions from all point sources with successful fits make up about 5% of the global total emissions in the HTAPv2 inventory. Future work should focus on reducing the uncertainties and applying similar methods to estimate

emissions for regions with a large number of sources in close proximity, or with large spatial extent (Fioletov et al., 2017), as well as emissions from fires (Mebust et al., 2011; Adams et al., 2019). Seasonal and diurnal emission cycles are currently one of the larger causes of uncertainty in satellite derived emissions. There is an overall lack of $NH_3$ emissions measurements, with only a few studies reporting flux measurements (Zöll et al., 2016; Schrader et al., 2018). Flux measurements for a larger number of locations and longer periods would be of great value to constrain diurnal emission cycles, which in turn will greatly reduce

the uncertainty of satellite based approaches. The second largest source of uncertainty is the bias in and between the CrIS and IASI products. Both satellite products have seen several improvements and/or upgrades (Van Damme et al., 2017; Shephard et al., 2019) and should be revalidated, both with FTIR-$NH_3$ (Dammers et al., 2016, 2017b) and/or ground and aircraft based





measurements (Van Damme et al., 2015). Furthermore, validation of the night-time $NH_3$ observations could enable use of the night time observations to constrain night time emissions and possibly lifetime of $NH_3$.

*Code and data availability.* The near-real-time IASI $NH_3$ (ANNI-$NH_3$-v2.1) and CO (FORLI) data used in this study is freely available through the AERIS database http://iasi.aeris-data.fr/NH3/ and http://iasi.aeris-data.fr/CO/. The CrIS data is currently available on request and
5 will be publicly released at the end of 2019. All Python code used to create any of the figures and/or to create the underlying data is available on request.

## Appendix A: IASI-A Mean total column map

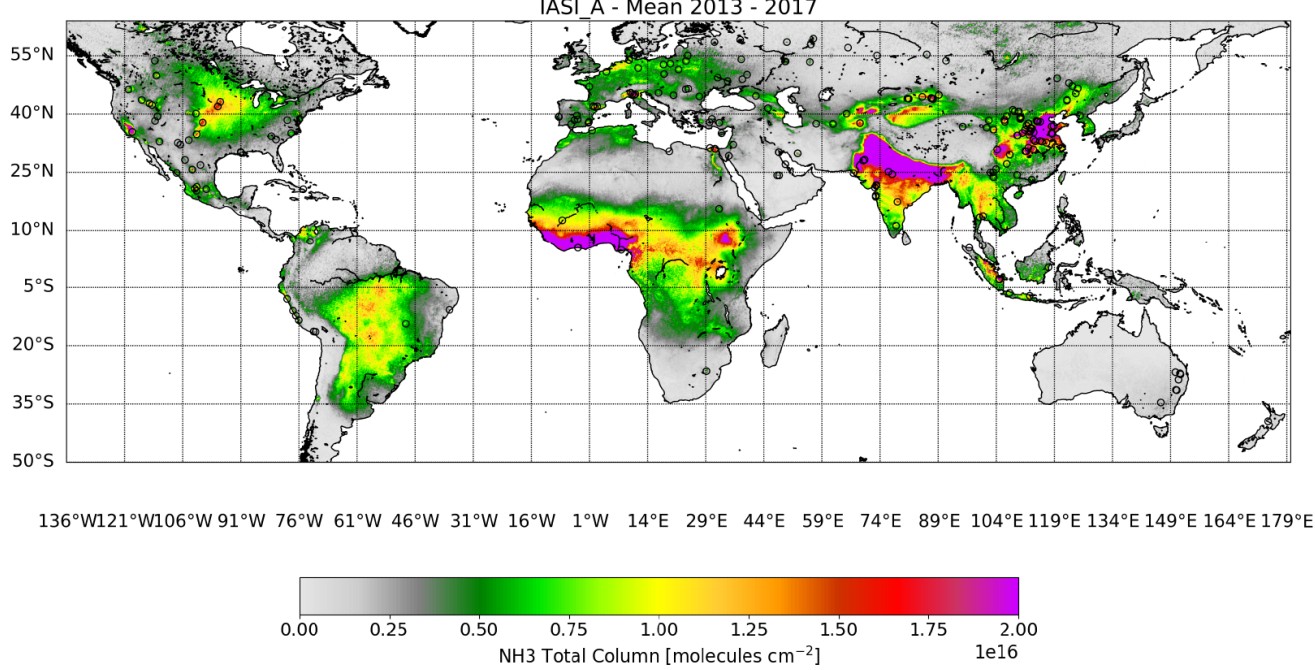

**Figure A1.** IASI-A $NH_3$ 5 year mean [2013-2017] total column [molecules cm$^{-2}$] distribution at 0.05° x 0.05° [longitude, latitude] resolution, The source locations with successful emission estimates are shown by black circles centered around the source locations.





## Appendix B:  The EMG fitting function

The function used in this study is a combination of an exponentially modified Gaussian (EMG) $g(y,s)$ and a Gaussian function $f(x,y)$ scaled by factor $a$ as shown in Eq. B1:

$$Column_{NH3}(x,y,s) = a \cdot f(x,y) \cdot g(y,s) + B \tag{B1}$$

$$f(x,y) = \frac{1}{\sigma_1\sqrt{2\pi}} exp\Big(-\frac{x^2}{2\sigma_1^2}\Big) \tag{B2}$$

$$g(y,s) = \frac{\lambda_1}{2} exp\Big(\frac{\lambda_1(\lambda_1\sigma^2 + 2y)}{2}\Big) erfc\Big(\frac{\lambda_1\sigma^2 + y}{\sqrt{2}\sigma}\Big) \tag{B3}$$

$$\sigma_1 = \begin{cases} \sqrt{\sigma^2 - 1.5y} & , y < 0 \\ \sigma & , y \geq 0 \end{cases} \tag{B4}$$

$$\lambda_1 = \frac{\lambda}{s} \tag{B5}$$

in which $x$, $y$ describe the crosswind and downwind location of the satellite pixel with respect to the source location [in km], $s$ [in km hr$^{-1}$] the wind speed, $\sigma$ the parameter describing the width of the Gaussian [in km], $\lambda$ the decay rate [hr$^{-1}$], $B$ the background total column concentration [molecules cm$^{-2}$], $a$ the emission enhancement [molecules cm$^{-2}$] and $erfc(x) = \frac{2}{\sqrt{\pi}} \int_x^\infty e^{-t^2} dt$. Equation B3, $g(y,s)$, is a convolution of a Gaussian describing the diffusion (with parameter $\sigma$ the width) that smooths an exponential function, which describes the exponential decay of the selected species in the downwind direction (with decay rate $\tau = 1/\lambda$, and $t = -y/s$ thus decay being $\sim exp(\lambda y/s)$ and $\lambda_1 = \lambda/s$, eq.B5). Equation B2, $f(x,y)$, describes the diffusion of our species perpendicular to the downwind direction. Similar to Fioletov et al. (2015) we adjusted the shape of the plume downwind to account for the size of the pixel shape and ensure convergence for a larger number of fits. Using a non-linear fitting algorithm we can fit the equation to the total column distribution and estimate $a$, $\sigma$ and $\lambda$. The fits are performed in Python using the non-linear curvefit package from the scipy module Jones et al. (2001) using the Levenberg-Marquant algorithm, which minimizes the difference between the given distribution and the fitted values. The fit is performed on a subset of the rotated grid, spanning 25 km in each crosswind direction and 36 km and 90 in the upwind and downwind directions. For our starting parameters we use; 15 km, 1/3 hr$^{-1}$, 1e19 molecules cm$^{-2}$ and 1e15 molecules cm$^{-2}$ for $\sigma$, $\lambda$, $a$ and $B$ respectively, representing an average of the $\sigma$ parameters found in Fioletov et al. (2015), an initial lifetime estimate of 3 hours, the initial enhancement estimate, and a low range estimate of the bias found in IASI and CrIS mean distributions. From a combination of $a$ and $\lambda$ we then find the emission rate $E = a \cdot \lambda$ (Fioletov et al., 2015).





## Appendix C: Results for all locations, free sigma and lifetime

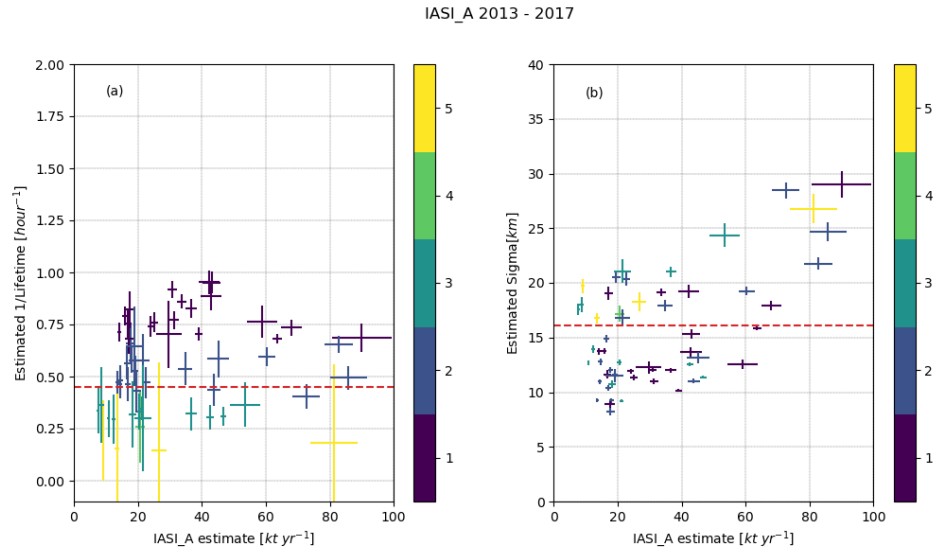

**Figure C1.** All successful fits for the 5-year (2013-2017) IASI-A dataset with a correlation > 0.5 and a SNR > 2. The left plot shows the fitted emissions versus the $\lambda$ parameter and the right side the fitted emissions vs the fitted $\sigma$ with the horizontal and vertical error bars showing the uncertainties of the fit. The color of the bars indicates the fitted lifetime rounded to the nearest integer. The horizontal striped red lines show the weighted average of the $\lambda$ and $\sigma$ parameters.

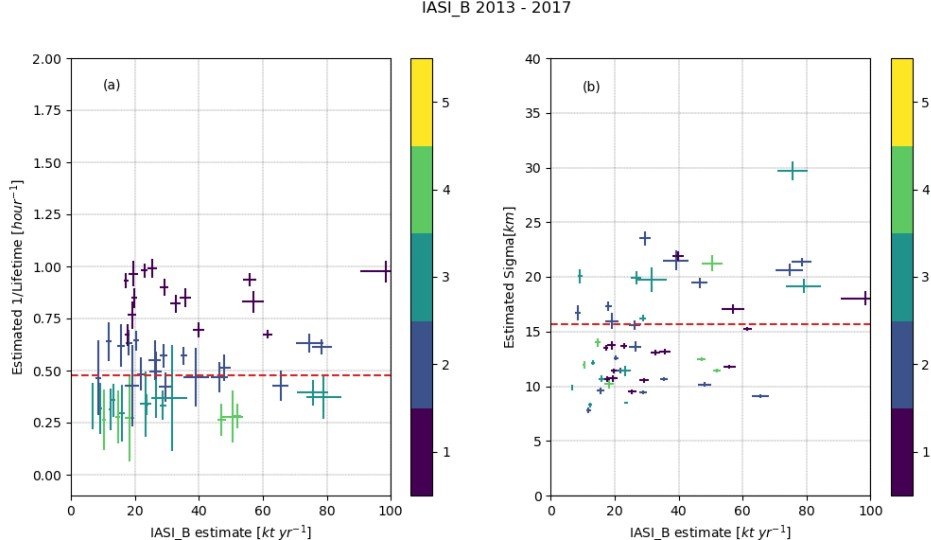

**Figure C2.** All successful fits for the 5-year (2013-2017) IASI-B dataset with a correlation > 0.5 and a SNR > 2. The left plot shows the fitted emissions versus the $\lambda$ parameter and the right side the fitted emissions vs the fitted $\sigma$ with the horizontal and vertical error bars showing the uncertainties of the fit. The color of the bars indicates the fitted lifetime rounded to the nearest integer. The horizontal striped red lines show the weighted average of the $\lambda$ and $\sigma$ parameters.

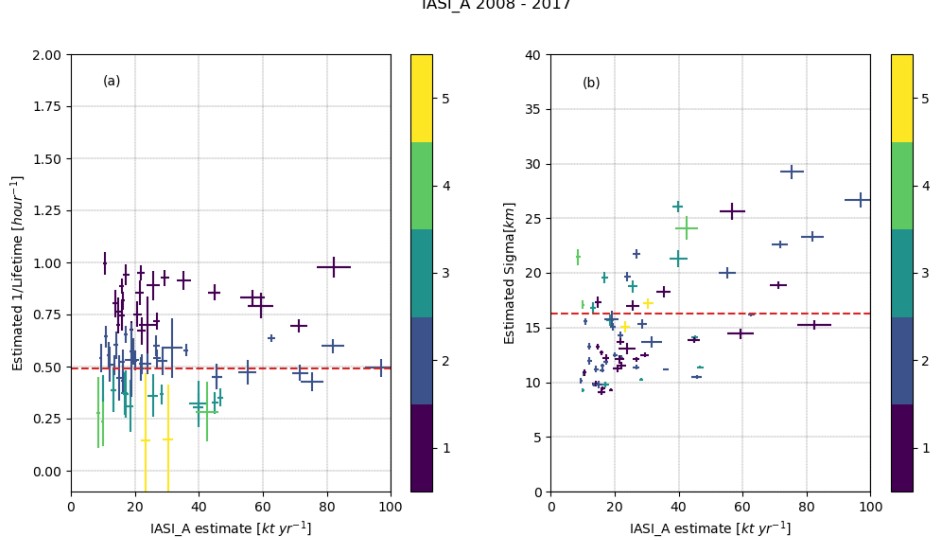

**Figure C3.** All successful fits for the 10-year (2008-2017) IASI-A dataset with a correlation > 0.5 and a SNR > 2. The left plot shows the fitted emissions versus the $\lambda$ parameter and the right side the fitted emissions vs the fitted $\sigma$ with the horizontal and vertical error bars showing the uncertainties of the fit. The color of the bars indicates the fitted lifetime rounded to the nearest integer. The horizontal striped red lines show the weighted average of the $\lambda$ and $\sigma$ parameters.





## Appendix D: Supporting material: Fit results for the NH₃ production plant near Shiraz, Iran.

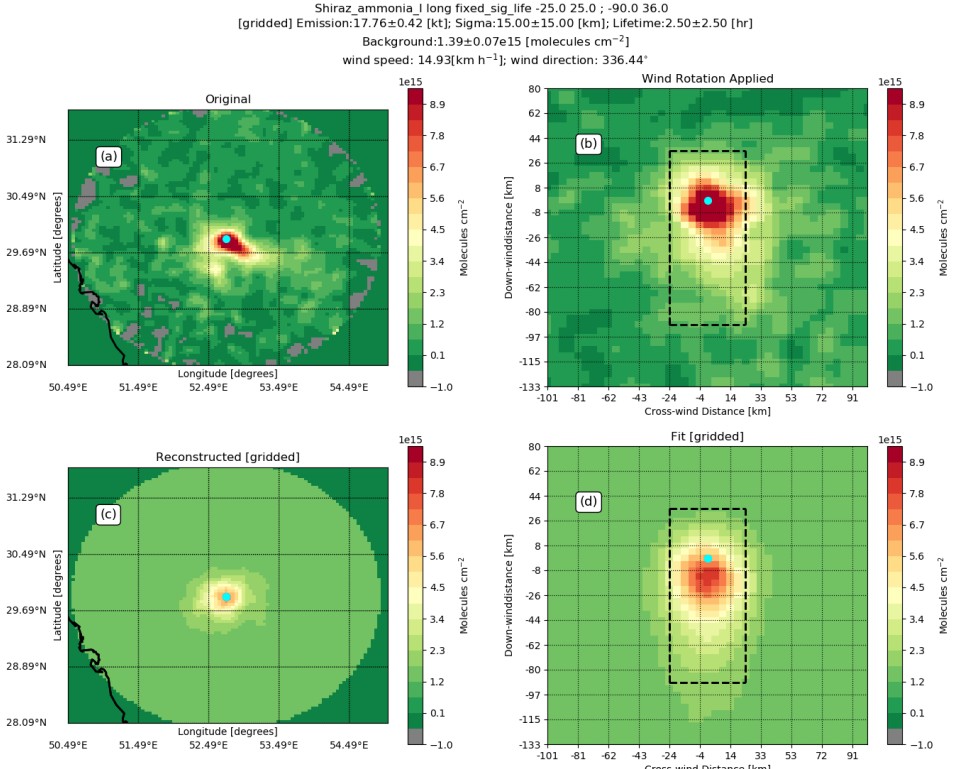

**Figure D1.** Fit example for the fertilizer plants in Shiraz, Iran. The top left plot shows the original IASI-A NH₃ total columns [2013-2017] gridded at a 0.05° x 0.04° [lon, lat] resolution with a 15 km oversampling length to represent the footprint. The top right plot shows the same observations after applying the wind rotation algorithm and gridding the observations to a 3 km x 4.5 km grid. The bottom right plot shows the fit to the observations and the bottom left plot the reconstruction of the initial plot using the parameters found in the fit.



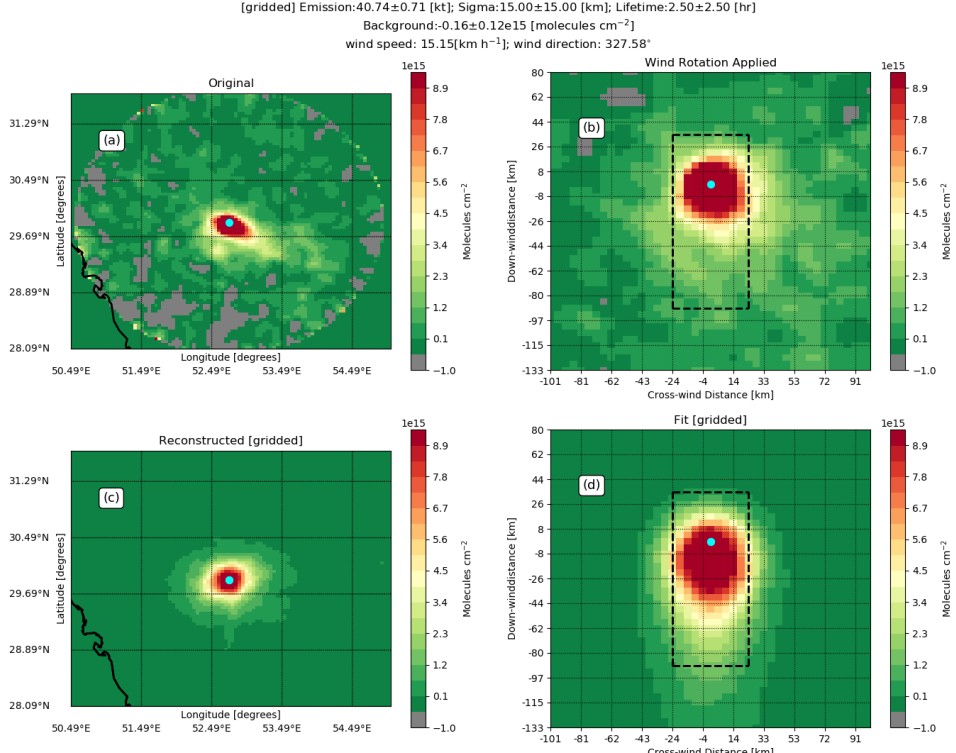

**Figure D2.** Fit example for the fertilizer plants in Shiraz, Iran. The top left plot shows the original IASI-B NH$_3$ total columns [2013-2017] gridded at a 0.05° x 0.04° [lon, lat] resolution with a 15 km oversampling length to represent the footprint. The top right plot shows the same observations after applying the wind rotation algorithm and gridding the observations to a 3 km x 4.5 km grid. The bottom right plot shows the fit to the observations and the bottom left plot the reconstruction of the initial plot using the parameters found in the fit.

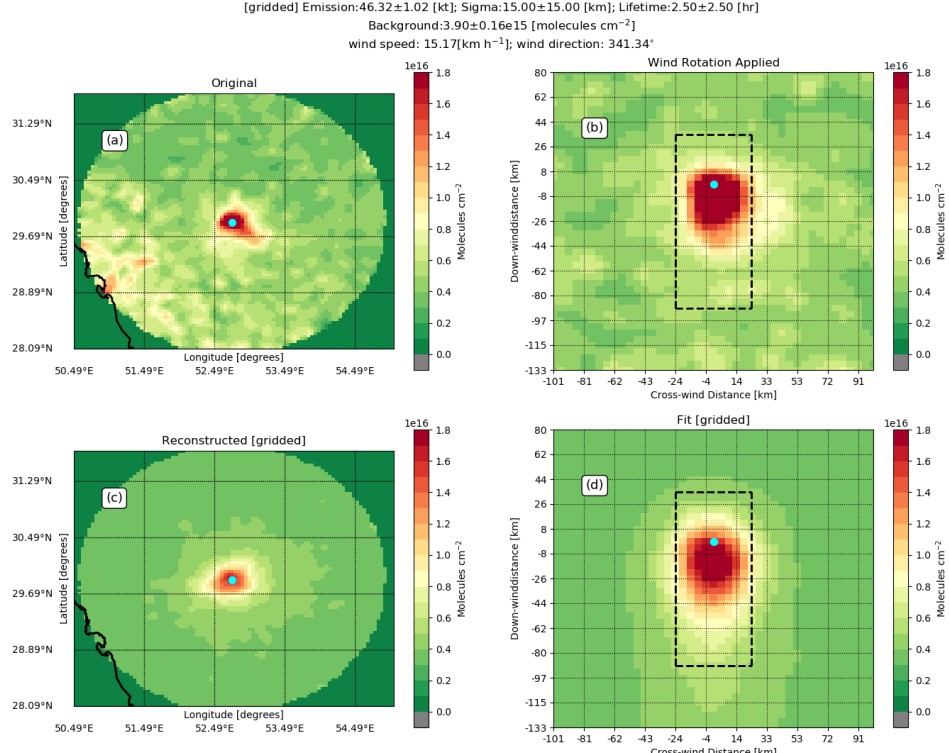

**Figure D3.** Fit example for the fertilizer plants in Shiraz, Iran. The top left plot shows the original CrIS NH$_3$ total columns [2013-2017] gridded at a 0.05° x 0.04° [lon, lat] resolution with a 15 km oversampling length to represent the footprint. The top right plot shows the same observations after applying the wind rotation algorithm and gridding the observations to a 3 km x 4.5 km grid. The bottom right plot shows the fit to the observations and the bottom left plot the reconstruction of the initial plot using the parameters found in the fit.





**Appendix E: Comparison to Van Damme et al. (2018)**

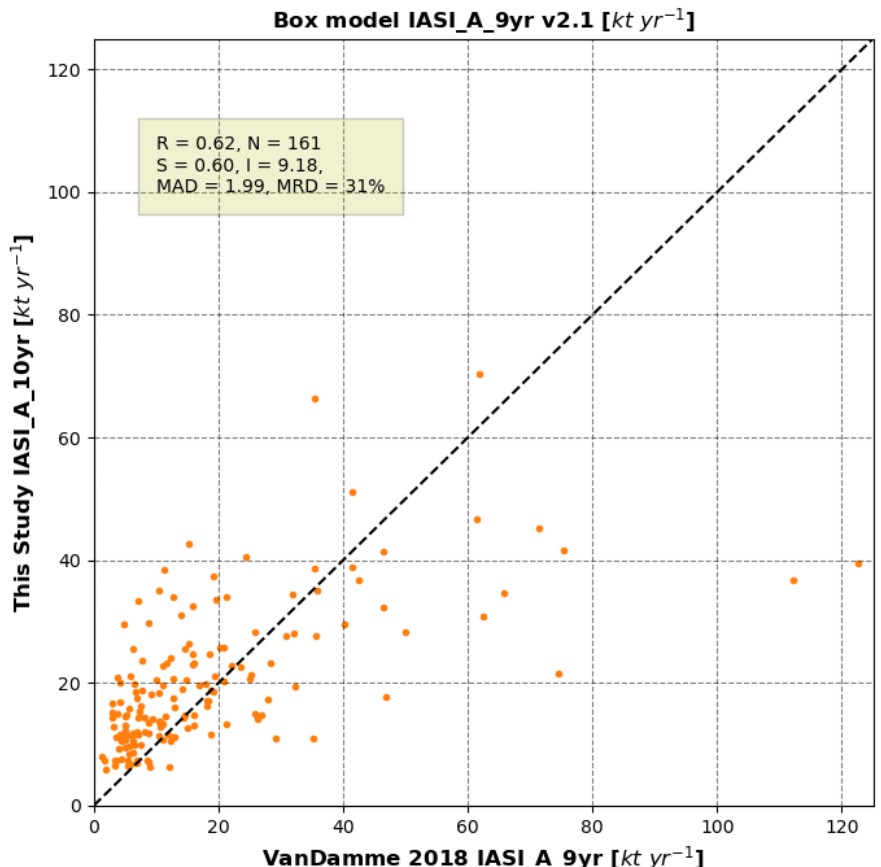

**Figure E1.** Comparison of emission estimates using this study's method versus those obtained using a box model approach of (Van Damme et al., 2018). The individual plots show the comparisons of this study, IASI-A (10yr), emission estimates, using a fixed sigma of 15 km and lifetime of 2.5 hours, versus the emissions estimates obtained using a box model approach on the 9 years (2008-2016) ANNI-NH$_3$-v2.1 IASI-A dataset. The later emission from Van Damme et al. (2018) have been adjusted to a lifetime of 2.5 hours and only locations characterized by a fit with a r>0.3 with the approach used in this study have been kept. R indicates the correlation between the sets, N the number of locations, S and I the resulting Slope and Intercept of the RMA regression and MAD and MRD the mean absolute difference (y-x) and the mean relative difference (y-x/(0.5y+0.5x)).

This study's results expand upon earlier work by Van Damme et al. (2018). The method used in this study is more advanced, with the addition of wind-rotation and a more complex treatment of the plume shape and transport, which enables the additional estimation of lifetime and plume spread ($\sigma$). For well-isolated point sources, the method has been shown to be more sensitive and accurate by McLinden et al. (2016) and de Foy et al. (2014) and has previously been successfully used to more accurately





estimate $SO_2$ emissions (Fioletov et al., 2015). Figure E1 shows the comparison of emission estimates using a box model approach (Van Damme et al., 2018) versus the results of this study. As Van Damme et al. (2018) used a mean lifetime of 12 for their reported results, we scale the estimates by a factor (12/2.5) to adjust the lifetime to the lifetime found in this study. In addition, they used only the 9 years of IASI-A observations available at that time, from a previous dataset (ANNI-$NH_3$-v2.1)

5    and the exact varying radius of the pixel footprints for the oversampling. Despite these differences, we find a correlation of r=0.62, with our study overall showing higher emissions with a relative difference of about 31%.

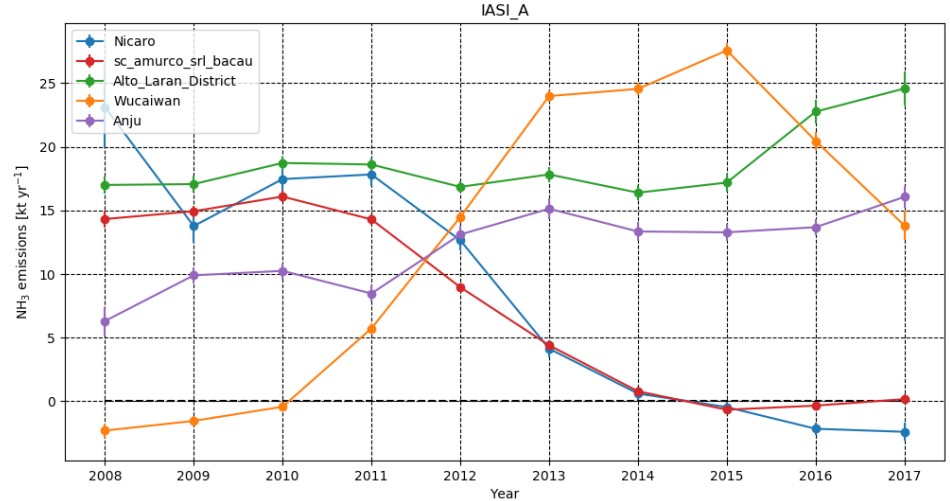

**Figure E2.** Example of IASI-A [2008-2017] three year running mean emissions for locations mentioned in Van Damme et al. (2018) using the method used in this study.





## Appendix F: Region dimensions

**Table F1.** Summary of regions mentioned in the results

| Region | Southern edge | Northern edge | Western edge | Eastern edge |
|---|---|---|---|---|
| **Europe** | 22.0° | 68.0° | -10.0° | 74.0° |
| **North America** | 15.0° | 64.0° | -136.0° | -55.0° |
| **India** | 5.0° | 39.0° | 65.0° | 100.0° |
| **China** | 18.0° | 58.0° | 72.0° | 143.0° |
| **Africa** | -40.0° | 40.0° | -30.0° | 60.0° |
| **Australia** | -50.0° | -5.0° | 110.0° | 180.0° |
| **South America** | -60.0° | 20.0° | -90.0° | -30.0° |
| **South East Asia** | -20.0° | 30.0° | 90.0° | 165.0° |

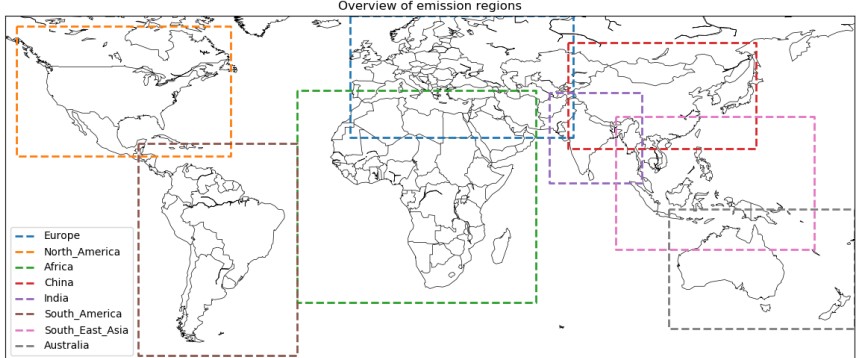

**Figure F1.** Overview of the emission regions.



*Author contributions.* ED, DG, and CML worked on the emission methods. MS, KCP, YGM and ED, worked on the CrIS retrieval. SvdG worked on the initial set of locations. MVD, SW, LC, PC produced the IASI-NH$_3$ dataset. CC produced the IASI-CO dataset. JWE, CML, MS, EL, VF, and MS helped interpret the results. The publication was prepared by ED, and all other authors contributed to the discussion of the paper.

5  *Competing interests.* The authors declare that they have no conflict of interest.

*Acknowledgements.* The CrIS input Level1 and Level2 data were obtained from the NOAA Comprehensive Large Array-Data Stewardship System (CLASS) (Liu et al., 2014), with special thanks to Axel Graumann (NOAA). The authors acknowledge the Aeris data infrastructure (https://iasi.aeris-data.fr/nh3/) for providing access to the IASI-NH$_3$ and IASI-CO data used in this study. We thank ECMWF for allowing free use of their meteorological fields. We acknowledge the use of the HTAPv2 dataset http://edgar.jrc.ec.europa.eu/htap_v2/. We thank Jacob
10  Hedelius and Mohit Manocha for the useful discussions.



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
