# Peer review of "NH3 emissions from large point sources derived from CrIS and IASI satellite observations"

_Atmospheric Chemistry and Physics, 2019_

## Referee Comment (RC1) · Anonymous Referee #1 · 12 Jul 2019

This paper is a very interesting analysis of NH3 point source emissions world-wide derived from the observations of CrIS and IASI. The manuscript is well written but sometimes the clarity can be improved, as I will specify in the comments below.

Page 2, line 24-25: This statement is rather vague. Can the authors be more quantitative by specifying how many are "a small number of locations"?

Page 3, line 8: Why is the underestimation of modelled concentrations likely due to underestimating emissions and not due to underestimating the lifetime?

Page 5, line 6: "have a nadir diameter of 14 km at nadir". The diameter is not in nadir direction.

Page 6, line 13-14: This sentence about oversampling is not very clear. Which data

is oversampled and how? The reference to Fioletov et al. (2011) does not mention oversampling.

Page 7, line 16-20: The regions Europe, Australia, Canada and the US are mentioned, but what has been used in other regions in the world? Please, clarify.

Page 7, line 23: "What do the authors mean with "up to several orders of uncertainty"? Please, specify the uncertainty.

Page 8, line 10: A 0.75 degree resolution is specified here as 40 km in both latitude and longitude direction. According to my calculation, 0.75 degree is about 75-80 km.

Page 8, line 14: The phrase "use 100 hPa of pressure layers at higher altitudes" is very unclear. I guess the authors mean that they use layers from the surface pressure to 100 hPa lower. Please, rephrase the sentence.

Page 9, line 14-15: If I look at equations B3 and B5, I conclude that the first part of the sentence should be "It tends to (underestimate) overestimate lifetime in...." to be in agreement with the statements in the rest of the sentence.

Page 10, line 14-15: This sentence is less ambiguous if it is written as "The fitted dataset in the longitude/latitude domain and in the downwind/crosswind domain are shown in Fig. 4(c) and (d), respectively.

Figure 3: I suggest removing all the text ("ZMU_ ...279.49") in the main title of this Figure.

Page 12, line 8: This sentence is missing a verb between "same model we" and "an uncertainty". found?

Page 13, line 2: "we first will first do free fits" => "we first will do free fits"

Page 13, line 30 and line 35: If the noise is 4 times lower why do the lower limits relate to each other with a factor 3?

Page 13, line 35: Which lower limit belongs to which instrument?

Page 13, line 35: What is the mentioned ratio? The ratio of what?

Table 1: I suggest putting a + in front of the 6.3 (diurnal var.), just for clarity.

Page 17, line 11: What is the rationale for the choice of 0.30 as a limit value?

Page 17, section 3.2: The main result of this paper, with all the nice information, is only available in a supplementary data file. Why not showing this in a Table in this section? It is maybe not feasible to show all the locations, but at least show a table with the 20 largest emitters.

Figure 10: I think the Figure can become clearer if all the symbols have the same size. I guess the size reflects the magnitude of emissions, but the colour is already clear enough for this.

Page 23, line3: "Only for the strongest sources is it was feasible to estimates using only ..". Please, correct all the typos in this sentence.

Page 27, section 4: the conclusions are clear, but I miss the fact mentioned that all results/conclusions are made for specific conditions: summertime and high wind speed.

Page 29, line18: tau is not the decay rate but the lifetime. Lambda is the decay rate.

Figure D1. It seems that the fit in this example is clearly lower than the measured total columns. Is that understood and part of the systematic uncertainties? Here I miss some discussion of the result, since this is one of your 4 selected example cases.

---

## Referee Comment (RC2) · Anonymous Referee #2 · 2 Aug 2019

NH3 emissions from large industrial and agricultural point sources are very uncertain, and are thought to have a minor contribution to the global total NH3 emissions. This manuscript has presented a pioneering study using satellite observations of ammonia (NH3) gas column concentrations from both CrIS and IASI to estimate the emission rates and lifetimes of NH3 from the large point sources over the world. The satellite derived NH3 emission estimates are compared with regional and global inventories, and show that current emission inventories generally miss or underestimate these point sources.

The manuscript is overall well organised and written. I recommend publish after the following comments been addressed.

[Figure]

**Specific Comments:**

(1) One comment is whether the satellite derived emission estimates can be influenced by the vertical sensitivities of the satellite measurements. For example, the CrIS-NH3 product is retrieved using the optimal estimation method. A priori profiles and averaging kernels matrices are then often required for comparing with other in-situ measurements. Satellite retrievals tend to have weak sensitivities in the boundary layer, and thus underestimate the true concentrations. Would it impact the results as presented in this study? Please discuss.

(2) Page 1, Line 13 in the abstract:
Please rewrite the sentence "which is equivalent to a factor of 2.5 between the CrIS estimated and HTAPv2 emissions". A factor of 2.5 compared with what values?

(3) Page 6, Line 9:
"only observations with a Quality Flag of 5". Please explain the meaning of "Quality Flag of 5" or list the reference.

(4) Page 8, Line 10:
"at a resolution of 0.75x0.75 resolution (40 x 40 km2)", 0.75 degree does not correspond to 40 km. Please check.

(5) Page 9, Line 9:
Please define sigma here in the text. Sigma is also used in Page 8, Line 30 with a different meaning.

(6) Page 10, Line 20:
Please also define lambda in the main text.

(7) Page 17, Figure 6:
For Figure 6, can you please explain why HTAP emission totals are integrated over
1 degree x 0.5 degree, rather than a finer resolution to compare with the point sources?

(8) Page 22, Table 4:
The Region total HTAPv2 value for China is too high due to the region define for China
(Figure F1) also covers the main NH3 emitting areas in the northern India. I suggest
add a table footnote to mention it.

(9) Page 29, Appendix B
Can you provide the range of fitted background concentrations (B)? Would high background NH3 concentrations over regions such as eastern China affect the applicability
of the fitting approach to estimate point emissions?

---

## Author Comment (AC1) · 19 Aug 2019

We would like to thank the referee for his/her time and insightful comments.

1. Ref1: Page 2, line 24-25: This statement is rather vague. Can the authors be more quantitative by specifying how many are "a small number of locations"?

1. Author: Only a few countries have instruments at more than five locations. Arguably the most densely covered is the Netherlands with both the MAN ($\sim$80 locations, Lolkema et al., 2015) and LML ($\sim$6 locations) networks. Of those sites, only $\sim$six instruments are in operation with measurement intervals better than 1 hour. The other sites are mostly monthly or bi-weekly averages. The other big networks can be found in the US (&Canada) and China. The AMON network in the US has about $\sim$60

sites that seem to regularly operate, which is poor compared to the total land surface area (http://nadp.slh.wisc.edu/AMoN/AMoNFactSheet.pdf). Quantitatively, if a number is needed, let us say that more than five locations in a country is rare. Even rarer is that sites are operated in a network, and have freely accessible data. Page 2, Line 25, Added "(>5 sites)" and added a reference (e.g. the MAN network, Lolkema et al., 2015) as an example.

2. Ref1: Page 3, line 8: Why is the underestimation of modelled concentrations likely due to underestimating emissions and not due to underestimating the lifetime?

2. Author: In a number of the referenced publications (for example Schiferl et al., 2014) the authors of several of the studies conclude that the emissions are likely to be underestimated. We agree that this is not true for all regions; therefore, we changed the word "likely" to "potentially".

3. Ref1: Page 5, line 6: "have a nadir diameter of 14 km at nadir". The diameter is not in nadir direction.

3. Author: Page 5, line 6: removed the first "nadir".

4. Ref1: Page 6, line 13-14: This sentence about oversampling is not very clear. Which data is oversampled and how? The reference to Fioletov et al. (2011) does not mention oversampling.

4. Author: Fioletov et al., 2011 mentions pixel averaging, see section[6], which follows the same steps as to what we now call oversampling. Fioletov et al., 2015, uses a similar approach. Page 6 Line 14: Changed the sentence to: "The original CrIS total column data has been smoothed by pixel averaging (Fioletov et al., 2011, 2015)" Figure 3 caption: changed the sentence to: "...the pixel averaging technique with a 15 km radius..." P33, Figure D1 caption: changed the sentence to: "...the pixel averaging technique with a 15 km radius..." P34, Figure D2 caption: changed the sentence to: "...the pixel averaging technique with a 15 km radius..." P35, Figure

D3 caption: changed the sentence to: "...the pixel averaging technique with a 15 km radius..." Page 37 line4: changed to: "...pixel footprints for the pixel averaging.

5. Ref1: Page 7, line 16-20: The regions Europe, Australia, Canada and the US are mentioned, but what has been used in other regions in the world? Please, clarify.

5. Author: In this study, we only used point source databases that were freely available. For the other regions, we used the HTAPv2 database instead. To our knowledge, only the four mentioned databases were freely available.

6. Ref1: Page 7, line 23: "What do the authors mean with "up to several orders of uncertainty"? Please, specify the uncertainty.

6. Author: We refer to the uncertainty ratings given by Kuenen et al., 2014. The rating range from A to E for different species and emission source categories. Road transport, industrial sources and disposal activities are classed under "E' and agriculture under "D", table 5 from Kuenen et al., 2014. This relates to a typical error range of 100%-300% for class D and "Order of Magnitude for "E". P7, L24: Changed the line to "that can have errors of an order of magnitude."

7. Ref1: Page 8, line 10: A 0.75 degree resolution is specified here as 40 km in both latitude and longitude direction. According to my calculation, 0.75 degree is about 75-80 km.

7. Author: P8, L12: Changed to"60x80 km2 at 45 degrees north".

8. Ref1: Page 8, line 14: The phrase "use 100 hPa of pressure layers at higher altitudes" is very unclear. I guess the authors mean that they use layers from the surface pressure to 100 hPa lower. Please, rephrase the sentence.

8. Author: P8 L15: "added: ($\sim$ 1000-900 hPa). P8 L15-17: Changed to ", For locations at higher altitudes we select layers that cover pressures from the surface pressure up to 100hPa less above the surface (for example 800-700 hPa, for a location with a surface pressure of 800 hPa).

9. Ref1: Page 9, line 14-15: If I look at equations B3 and B5, I conclude that the first part of the sentence should be "It tends to (underestimate) overestimate lifetime in...." to be in agreement with the statements in the rest of the sentence.

9. Author: P10, L2-3: Changed to "It tends to (underestimate) overestimate lifetime in..."

10. Ref1: Page 10, line 14-15: This sentence is less ambiguous if it is written as "The fitted dataset in the longitude/latitude domain and in the downwind/crosswind domain are shown in Fig. 4(c) and (d), respectively.

10. Author: P10, L17: Adapted the suggested sentence.

11. Ref1: Figure 3: I suggest removing all the text ("ZMU_ ...279.49") in the main title of this Figure.

11. Author: Adjusted Figure 3 as suggested.

12. Ref1: Page 12, line 8: This sentence is missing a verb between "same model we" and "an uncertainty". found?

12. Author: P12, L9: Added, "calculated"

13. Ref1: Page 13, line 2: "we first will first do free fits" => "we first will do free fits"

13. Author: P13, L4: Adjusted to: "In this study we first will do free fits,"

14. Ref1: Page 13, line 30 and line 35: If the noise is 4 times lower why do the lower limits relate to each other with a factor 3?

14. Author: The CrIS and IASI products use different approaches for the retrieval. Therefore, we do not expect a 1:1 conversion of noise levels to uncertainties. It is beyond the scope of this manuscript to go into detail on why the reported uncertainty limits do not relate 1:1 with the reported noise levels.

15. Ref1: Page 13, line 35: Which lower limit belongs to which instrument?

15. Author: P14, line 2: Added "for IASI" and "for CrIS".

16. Ref1: Page 13, line 35: What is the mentioned ratio? The ratio of what?

16. Author: P14, line 3: Changed to" The ratio between surface concentration (ppb) and total column density is highly variable"

17. Ref1: Table 1: I suggest putting a + in front of the 6.3 (diurnal var.), just for clarity.

17. Author: Added as suggested to Table 1.

18. Ref1: Page 17, line 11: What is the rationale for the choice of 0.30 as a limit value?

18. Author: For the free fits, we used a stricter r>0.5 limit. For the fits with the fixed parameters, the individual fits will become worse. Therefore, we decided to use a looser value of 0.3. We have added the correlation value to the datasets in the supplementary. If any of the readers wants to use a stricter limit, they can apply it. P17, L10-12: Changed lines to: "The results of those fits are quality controlled as before. . ... but now with a weaker correlation filter (r<0.30) to adjust for the stricter fit parameters. All results, including the correlations, are merged into the location list and can be found in the supplementary material."

19. Ref1: Page 17, section 3.2: The main result of this paper, with all the nice information, is only available in a supplementary data file. Why not showing this in a Table in this section? It is maybe not feasible to show all the locations, but at least show a table with the 20 largest emitters.

19. Author: We contemplated to put the table with emissions in the appendix; however, the number of sources and amount ancillary data was too much to fit properly in a limited number of pages. The four later examples in the text (Figure 6, and page 17) + initial example of the ZMU fertilizer plant (page 10 + Figure 3.) give ample information about the method and capabilities. The science in the manuscript is not just about reporting the largest emitters, but also that it is possible to detect smaller sources. With the large number of source locations, it is best to provide this information in a

more supplemental database format for users.

20. Ref1: Figure 10: I think the Figure can become clearer if all the symbols have the same size. I guess the size reflects the magnitude of emissions, but the colour is already clear enough for this.

20. Author: Changed as suggested.

21. Ref1: Page 23, line3: "Only for the strongest sources is it was feasible to estimates using only ..." Please, correct all the typos in this sentence.

21. Author: Changed to" Estimates using only a single year of observations were only possible for the strongest sources."

22. Ref1: Page 27, section 4: the conclusions are clear, but I miss the fact mentioned that all results/ conclusions are made for specific conditions: summertime and high wind speed.

22. Author: The results are applicable for summertime conditions and wind speeds above 0.5 km h-1, with lower uncertainties for sites with average to high wind speeds 10-20 km h-1. P27, L26-29: Added in "The estimates are based on the spring and summertime observations and are representative of those six months. The yearly emission total follows from the assumption of constant emission cycle throughout the year, which can lead to an overestimation for sources with a strong seasonal cycle."

23. Ref1: Page 29, line18: tau is not the decay rate but the lifetime. Lambda is the decay rate.

23. Author: Adjusted to "lifetime" as suggested.

24. Ref1: Figure D1. It seems that the fit in this example is clearly lower than the measured total columns. Is that understood and part of the systematic uncertainties? Here I miss some discussion of the result, since this is one of your 4 selected example cases.

24. Author: This is our example of a non-ideal location, which we mentioned in the text. The deviation is attributed to the orography and very low mean wind speeds around the site. See page 18, line 13-16. "The fits for NH3 production plants near Shiraz . . . limited applicability ... large elevation changes ... a single outflow direction. While the fits may seem reasonable . . .the overall wind speed is very low and the plume shape is not well defined."

References: Fioletov, V., McLinden, C., Krotkov, N., Moran, M., and Yang, K.: Estimation of SO2 emissions using OMI retrievals, Geophysical Research Letters, 38, 2011.

Fioletov, V., McLinden, C., Krotkov, N., and Li, C.: Lifetimes and emissions of SO2 from point sources estimated from OMI, Geophysical Research Letters, 42, 1969–1976, 2015.

Lolkema, D. E., Noordijk, H., Stolk, A. P., Hoogerbrugge, R., van Zanten, M. C., and van Pul, W. A. J.: The Measuring Ammonia in Nature (MAN) network in the Netherlands, Biogeosciences, 12, 5133-5142, https://doi.org/10.5194/bg-12-5133-2015, 2015.

Kuenen, J. J. P., Visschedijk, A. J. H., Jozwicka, M., and Denier van der Gon, H. A. C.: TNO-MACCII emission inventory; a multi-year (2003-2009) consistent high-resolution European emission inventory for air quality modelling, Atmospheric Chemistry and Physics, 14, 10 963–10 976, https://doi.org/10.5194/acp-14-10963-2014, https://www.atmos-chem-phys.net/14/10963/2014/, 2014.

───────────────────────────

---

## Author Comment (AC2) · 19 Aug 2019

We would like to thank the referee for his/her time and insightful comments.

(1) Ref2: One comment is whether the satellite derived emission estimates can be influenced by the vertical sensitivities of the satellite measurements. For example, the CrIS-NH3 product is retrieved using the optimal estimation method. A priori profiles and averaging kernels matrices are then often required for comparing with other in-situ measurements. Satellite retrievals tend to have weak sensitivities in the boundary layer, and thus underestimate the true concentrations. Would it impact the results as presented in this study? Please discuss.

(1) Author: We agree with the review and that is one of the reasons why we use a

2d plume model in combination with the total column densities, which reduces the dependency on the vertical sensitivity. The comparison of the CrIS-NH3 with FTIR-NH3 observations does not show any systematic under or over estimation in the total column; see page 4 lines 11-13. The most recent FTIR validation of the previous IASI product does show an underestimation. For reference in the text, see page 6 lines 31-34.

(2) Ref2: Page 1, Line 13 in the abstract: Please rewrite the sentence "which is equivalent to a factor of 2.5 between the CrIS estimated and HTAPv2 emissions". A factor of 2.5 compared with what values?

(2) Author: Page 1, Line 12-14: Change the sentence to "The CrIS emission estimates give a total of 5622 kt yr-1, for the sources analyzed in this study, which is around a factor 2.5 higher than the emissions reported in HTAPv2."

(3) Ref2: Page 6, Line 9: "only observations with a Quality Flag of 5". Please explain the meaning of "Quality Flag of 5" or list the reference.

(3) Author: Page 6, Line 10: added "(Shephard et al., 2019)". It reflects observations with a maximum NH3 concentration of 200 ppb for single layers in the profile. A DOF >0.1. Spectral signal-Noise-Ratio of >1. A thermal contrast above 0. In addition, a max CHI2 of the optimal estimation fit of 5.0. All the other flags are defined in Shephard et al., 2019.

(4) Ref2: Page 8, Line 10: "at a resolution of 0.75x0.75 resolution (40 x 40 km2)", 0.75 degree does not correspond to 40 km. Please check.

(4) Author: Page 8, Line 12: changed to"60x80 km2 at 45 degrees north"

(5) Ref2: Page 9, Line 13: Please define sigma here in the text. Sigma is also used in Page 8, Line 30 with a different meaning.

(5) Author: Changed the sigma on page 8, line 34 to "standard deviations". Added "the plume spread ($\sigma$) to line 9.

(6) Ref2: Page 10, Line 20: Please also define lambda in the main text.

(6) Author: Page 9, line 12-13: Added "($\tau = 1/\lambda$, with $\lambda$ the decay rate)".

(7) Ref2: Page 17, Figure 6: For Figure 6, can you please explain why HTAP emission totals are integrated over 1 degree x 0.5 degree, rather than a finer resolution to compare with the point sources?

(7) Author: The HTAP emission totals include locations with agricultural emissions. Those locations are slightly less point source like and therefore we widened the integration area to capture the total source.

(8) Ref2: Page 22, Table 4: The Region total HTAPv2 value for China is too high due to the region define for China (Figure F1) also covers the main NH3 emitting areas in the northern India. I suggest add a table footnote to mention it.

(8) Author: Added a footnote to Table 4, "*The China region includes a large part of northern India, therefore the emissions may seem higher than expected."

(9) Ref2: Page 29, Appendix B Can you provide the range of fitted background concentrations (B)? Would high background NH3 concentrations over regions such as eastern China affect the applicability of the fitting approach to estimate point emissions?

(9) Author: The fitted background concentrations depend both on satellite and the region with higher background values for regions with other sources, and generally higher background values for CrIS compared to IASI-A and –B. As long as the background concentrations are homogeneous, there should be no effect to the emission estimate. This is where the SNR value comes in, which we use to determine if most of the "information" is coming from our intended source. We filter for an SNR>2, to ensure that the variations near the source are higher than those further upwind of the source. Sources very near (<10km) the source will however be included in the final emission estimate as it is not possible to discern these from the intended source, with the single source approach used in this paper. We marked the sites with nearby agricultural sources

in Fig 9 to show the potential interference. Furthermore, we added both the HTAPv2 emissions with and without agriculture in the supplementary tables.

References:

Shephard, M. W., Dammers, E., Kharol, S., and Cady-Pereira, K.: Ammonia measurements from space with the Cross-track Infrared Sounder (CrIS): characteristics and applications, in preparation for ACP, 2019

―――――――――――――――――